# The Calpain-7 protease functions together with the ESCRT-III protein IST1 within the midbody to regulate the timing and completion of abscission

Elliott L Paine[1], Jack J Skalicky[1], Frank G Whitby[1], Douglas R Mackay[2], Katharine S Ullman[2], Christopher P Hill[1], Wesley I Sundquist[1]*

[1]Department of Biochemistry, University of Utah School of Medicine, Salt Lake City, United States; [2]Department of Oncological Sciences, Huntsman Cancer Institute, University of Utah, Salt Lake City, United States

**Abstract** The Endosomal Sorting Complexes Required for Transport (ESCRT) machinery mediates the membrane fission step that completes cytokinetic abscission and separates dividing cells. Filaments composed of ESCRT-III subunits constrict membranes of the intercellular bridge midbody to the abscission point. These filaments also bind and recruit cofactors whose activities help execute abscission and/or delay abscission timing in response to mitotic errors via the NoCut/Abscission checkpoint. We previously showed that the ESCRT-III subunit IST1 binds the cysteine protease Calpain-7 (CAPN7) and that CAPN7 is required for both efficient abscission and NoCut checkpoint maintenance (Wenzel et al., 2022). Here, we report biochemical and crystallographic studies showing that the tandem microtubule-interacting and trafficking (MIT) domains of CAPN7 bind simultaneously to two distinct IST1 MIT interaction motifs. Structure-guided point mutations in either CAPN7 MIT domain disrupted IST1 binding in vitro and in cells, and depletion/rescue experiments showed that the CAPN7-IST1 interaction is required for (1) CAPN7 recruitment to midbodies, (2) efficient abscission, and (3) NoCut checkpoint arrest. CAPN7 proteolytic activity is also required for abscission and checkpoint maintenance. Hence, IST1 recruits CAPN7 to midbodies, where its proteolytic activity is required to regulate and complete abscission.

*For correspondence:
wes@biochem.utah.edu

## Editor's evaluation

This fundamental study provides insight into the role of the Calpain-7 protease and its proteolytic activity in cell division. Using rigorous molecular and cellular approaches, they provide compelling evidence for the role of the protease in the timing and completion of cell abscission. Conclusions are supported with strong mutagenesis and rescue assays. The work will be of broad interest to cell biologists and biochemists.

## Introduction

Midbody abscission separates two dividing cells at the end of cytokinesis. The Endosomal Sorting Complexes Required for Transport (ESCRT) pathway is central to abscission and to its regulation via the NoCut/Abscission checkpoint, whereby abscission delay allows mitotic errors to be resolved or protected (*Carlton and Martin-Serrano, 2007*; *Morita et al., 2007*; *Carlton et al., 2012*; *Capalbo et al., 2012*; *Capalbo et al., 2016*; *Scourfield and Martin-Serrano, 2017*). Humans express 12 distinct ESCRT-III proteins that are recruited to membrane fission sites throughout the cell, including

to midbodies that connect dividing cells during cytokinesis. Within the midbody, ESCRT-III proteins copolymerize into filaments that constrict the membrane to the fission point (*Guizetti et al., 2011*; *Elia et al., 2011*; *Mierzwa et al., 2017*; *Nguyen et al., 2020*; *Pfitzner et al., 2021*; *Azad et al., 2022*). ESCRT-III filaments also recruit a variety of cofactors, including the VPS4 AAA+ ATPases, which dynamically remodel the filaments to drive midbody constriction (*Elia et al., 2012*; *Mierzwa et al., 2017*; *Pfitzner et al., 2020*; *Pfitzner et al., 2021*). Many of these cofactors contain microtubule-interacting and trafficking (MIT) domains, which bind differentially to MIT-interacting motifs (MIMs) located near the C termini of the different human ESCRT-III proteins (*Hurley and Yang, 2008*; *Wenzel et al., 2022*).

Our recent quantitative survey of the human ESCRT-III-MIT interactome revealed a series of novel interactions between ESCRT-III subunits and MIT cofactors, and implicated a subset of these cofactors in abscission and NoCut checkpoint maintenance (*Wenzel et al., 2022*). One such cofactor was Calpain-7 (CAPN7), a ubiquitously expressed but poorly understood cysteine protease that had not previously been linked to abscission or to the NoCut checkpoint. CAPN7 contains tandem MIT domains that can bind specifically to the ESCRT-III subunit IST1 (*Osako et al., 2010*; *Wenzel et al., 2022*). This interaction can activate CAPN7 autolysis and proteolytic activity towards non-physiological substrates (*Osako et al., 2010*; *Maemoto et al., 2013*), although authentic CAPN7 substrates are not yet known. We undertook the current studies with the goals of defining precisely how CAPN7 and IST1 interact, how CAPN7 is recruited to midbodies, and whether CAPN7 must function as a midbody protease in order to support efficient abscission and maintain NoCut checkpoint signaling.

## Results and discussion
### The CAPN7 MIT domains bind distinct IST1 MIM elements

The tandem CAPN7 MIT domains (CAPN7(MIT)$_2$) bind a C-terminal region of IST1 that contains two distinct MIM elements (*Figure 1A*; *Agromayor et al., 2009*; *Bajorek et al., 2009*; *Osako et al., 2010*; *Wenzel et al., 2022*). We quantified this interaction using fluorescence polarization anisotropy (FPA) binding assays with purified, recombinant CAPN7(MIT)$_2$ and fluorescently labeled IST constructs that spanned either one or both of the MIM elements. These experiments showed that both IST1 MIMs contribute to binding (*Figure 1B*; *Osako et al., 2010*; *Wenzel et al., 2022*). The double MIM IST1$_{316-366}$ construct bound CAPN7(MIT)$_2$ tightly (K$_D$ = 0.09 ± 0.01 μM), with the C-terminal IST1 MIM element (IST1$_{344-366}$) contributing most of the binding energy (K$_D$ = 1.8 ± 0.1 μM). The N-terminal IST1 MIM element (MIM$_{316-343}$) bound too weakly to measure accurately by FPA (K$_D$ > 100 μM), so NMR titration experiments were performed to confirm the specificity and quantify the binding energy (K$_D$ = 200 ± 20 μM) (*Figure 1—figure supplement 1*). These experiments demonstrated that both IST1 MIM elements contribute to binding CAPN7(MIT)$_2$, in good agreement with previous measurements (*Wenzel et al., 2022*).

Binding-dependent changes in amide resonance intensities within IST1$_{303-366}$ were also used to identify IST1 residues that bind CAPN7(MIT)$_2$ and guide the design of a minimal IST1 peptide for structural studies. Our NMR titration experiments utilized a fully assigned $^{15}$N-labeled IST1$_{303-366}$ peptide spanning both MIM elements (*Caballe et al., 2015*) and unlabeled CAPN7(MIT)$_2$. Peak intensity ratios (unbound/bound) were measured for all 55 main chain amide resonances (*Figure 1C and D* and *Figure 1—figure supplement 2*). 20/55 IST1$_{303-366}$ amide resonances displayed large (>15-fold) changes in peak intensity upon saturation binding of CAPN7(MIT)$_2$. These residues mapped exclusively to the two known IST1 MIMs (*Figure 1D*), again implicating both elements in CAPN7(MIT)$_2$ binding and implying that residues outside of these elements do not contribute to binding. Consistent with this idea, a minimal IST1 construct (IST1$_{322-366}$, K$_D$ = 0.13 ± 0.01 μM) bound with the same affinity as the original construct (IST1$_{316-366}$, K$_D$ = 0.11 ± 0.01 μM) (*Figure 1—figure supplement 3*). In summary, our experiments demonstrated that both IST1 MIM elements contribute to CAPN7 binding and defined IST1$_{322-366}$ as a minimal CAPN7(MIT)$_2$ binding construct that was used in subsequent structural studies.

### Crystal structure of CAPN7(MIT)$_2$–IST1 MIMs complex

We determined the crystal structure of the CAPN7(MIT)$_2$–IST1$_{322-366}$ complex to define the interaction in molecular detail and distinguish between the four possible binding modes in which individual CAPN7 MIT domains either bound simultaneously to both IST1 MIMs, as reported in other MIT:MIM

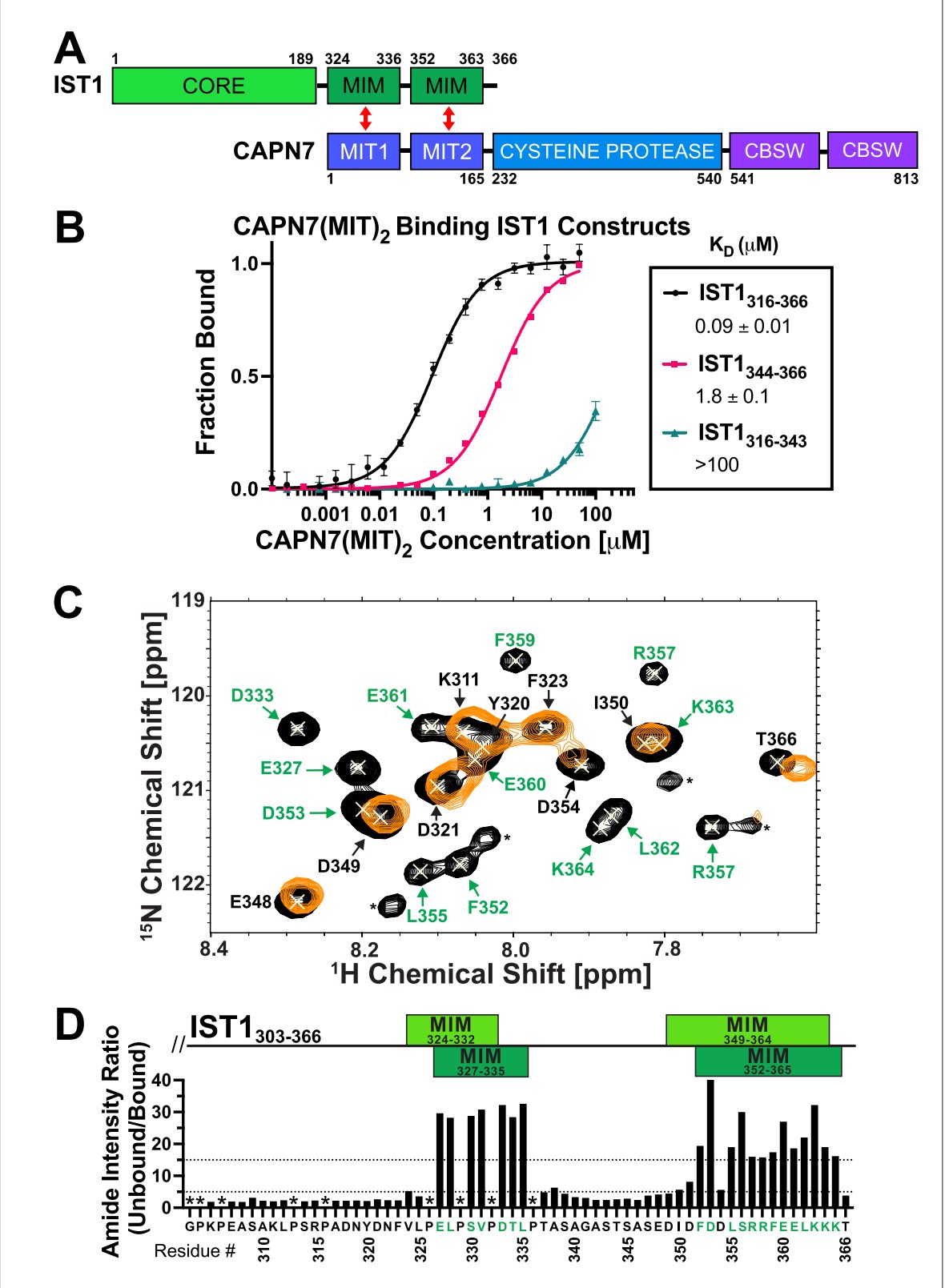

**Figure 1.** CAPN7 binds IST1 through tandem microtubule-interacting and trafficking (MIT) domains. (**A**) Domain organization of CAPN7 and IST1, depicting the binding interaction between tandem MIT domains of CAPN7 and MIT-interacting motif (MIM) elements of IST1 (red double-headed arrows). Domain definitions: CORE, helical ESCRT-III core domain of IST1 that functions in filament formation; CBSW, tandem calpain-type beta sandwich domains of CAPN7. (**B**) Fluorescence polarization anisotropy binding isotherms showing CAPN7(MIT)$_2$ binding to IST1 constructs spanning

*Figure 1 continued on next page*

*Figure 1 continued*

tandem or individual MIM elements (IST1$_{316-366}$, IST1$_{316-343}$, and IST1$_{344-366}$, respectively). Isotherm data points and dissociation constants (K$_D$) are averages ± standard error of the mean from three independent experiments. Error bars on the IST1$_{344-366}$ isotherm are entirely masked by the data symbols. (**C**) NMR mapping of the CAPN7(MIT)$_2$ binding sites on IST1$_{303-366}$. Sections of overlaid HSQC spectra of free IST1$_{303-366}$ (black contours) and IST1$_{303-366}$ saturated with 1.3 molar equivalents of CAPN7(MIT)$_2$ (orange contours) are shown. Amide NH resonances in the unbound state (black contours) that lack bound state resonances (orange contours) correspond to residues that experience large intensity perturbations upon CAPN7 binding (amino acid residue labels in green). In contrast, strong resonances that overlap well in both the unbound and bound states (black and orange contours) correspond to residues that experience smaller intensity perturbations upon CAPN7 binding (amino acid residue labels in black) (see *Figure 1—figure supplement 2* for the entire spectra). (**D**) Amide intensity ratios (unbound/bound) for each residue of the IST1$_{303-366}$ peptide. Small ratios (<5, 30 residues, lower dotted line) correspond to residues that remain dynamic in the complex, whereas large ratios (>15, 20 residues, upper dotted line) correspond to residues whose dynamics are reduced upon complex formation (and therefore likely contact CAPN7(MIT)$_2$ and/or become ordered upon binding). Proline residues were not scored (asterisks). IST1 MIM elements show either the bounds of interpretable electron density from the crystal structure of the complex (top boxes, light green, see *Figure 2* and *Figure 2—figure supplement 1*) or the bounds of the complex as defined by NMR resonance intensity changes (bottom boxes, dark green, see (**C**) and *Figure 1—figure supplement 2*).

The online version of this article includes the following figure supplement(s) for figure 1:

**Figure supplement 1.** NMR titration of $^{15}$N-labeled IST1$_{314-343}$ with unlabeled CAPN7$_{1-75}$.

**Figure supplement 2.** NMR spectra of free and CAPN7(MIT)$_2$-bound $^{15}$N-labeled IST1$_{303-366}$.

**Figure supplement 3.** CAPN7(MIT)$_2$ binds equally well to IST1$_{316-366}$ and the minimal IST1$_{322-366}$ construct.

**Figure supplement 4.** Raw fluorescence polarization anisotropy binding isotherms, best-fit models, and associated statistics corresponding to the normalized binding isotherms presented in *Figure 1B*.

interactions (*Bajorek et al., 2009*; *Wenzel et al., 2022*), or each CAPN7 MIT domain bound a different IST1 MIM element in one of two possible pairwise interactions. The structure revealed that each MIT domain binds a different MIM element, which is a configuration that has not been documented previously (*Figure 2A* and *Table 1*). The asymmetric unit contains two copies of the CAPN7(MIT)$_2$–IST1$_{322-366}$ complex, and the flexible linkers between the different MIT and MIM elements could, in principle, connect the linked IST1 MIM-CAPN7 complexes in two different ways. We have illustrated the simpler of the two models, which minimizes polypeptide chain cross-overs, but either connection is physically possible and the choice of connectivity does not affect our interpretation of the model.

Both CAPN7 MIT domains form the characteristic three-helix MIT bundle structure (*Scott et al., 2005*), and they are connected by a disordered six-residue linker. The N-terminal IST1 MIM element binds in an extended conformation in the helix 1/3 groove of the N-terminal CAPN7 MIT domain, making a canonical 'type 2' (MIM2) interaction (*Figure 2A*, top, and *Figure 2—figure supplement 1A*; *Kieffer et al., 2008*; *Samson et al., 2008*; *Vild and Xu, 2014*; *Kojima et al., 2016*; *Wenzel et al., 2022*). The binding interface spans IST1 residues 324–332 ($_{324}$VLPELPSVP$_{332}$), which resembles the previously defined MIM2 consensus sequence ($\Phi$PX$\Phi$PXXP$\Phi$P, where $\Phi$ represents a hydrophobic residue and X a variable (often polar) residue; *Kojima et al., 2016*). Consistent with this observation, the overall structure closely resembles the type 2 ESCRT-III-MIT complex formed by CHMP6$_{168-179}$ bound to the VPS4A MIT domain (*Figure 2B* and *Figure 2—figure supplement 2*; *Kieffer et al., 2008*; *Skalicky et al., 2012*).

Sequence divergence at either end of the MIM core appears to explain why the MIM2 element of CHMP6 binds preferentially to the VPS4A MIT domain, whereas the MIM2 element of IST1 prefers the first MIT domain of CAPN7 (*Figure 2—figure supplement 2A*). In the CAPN7-IST1 complex, IST1 Ser330 bulges away from the MIT domain to accommodate an intramolecular salt bridge formed by CAPN7 Asp21 and Arg63 (*Figure 2—figure supplement 2A and C*), and the peptide then bends back to allow Val331 and Pro332 to make hydrogen bonds and hydrophobic contacts, respectively. In the VPS4A MIT-CHMP6 complex, the CHMP6 Glu176 residue forms a salt bridge with VPS4A Lys23, whereas the equivalent interaction between CAPN7 MIT Gly24 and IST1 Val331 interaction has a very different character, thereby disfavoring CHMP6 binding to CAPN7 (*Figure 2—figure supplement 2A and C*). At the other end of the interface, helix 3 of the VPS4A MIT domain is two turns longer than the equivalent helix 3 in CAPN7 MIT (*Figure 2B*). This allows CHMP6 Ile168 to make a favorable interaction, whereas the shorter CAPN7 MIT helix 3 projects loop residue Leu50 directly toward the IST1$_{324-332}$ N-terminus, in a position that would disfavor CHMP6 Ile168 binding (*Figure 2—figure supplement 2A and B*).

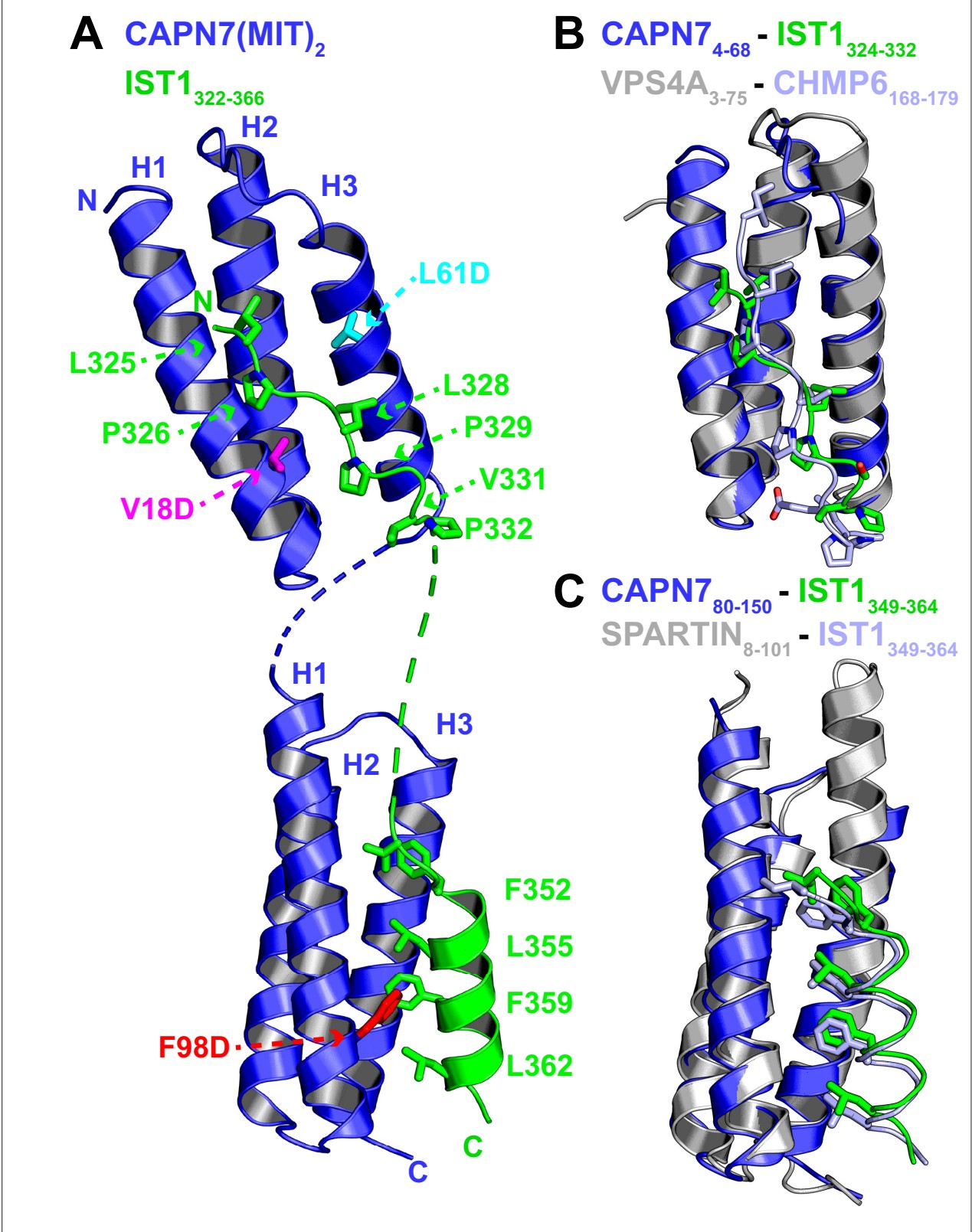

**Figure 2.** Crystal structure of the CAPN7(MIT)₂-IST1₃₂₂₋₃₆₆ complex. (**A**) Ribbon representation of CAPN7(MIT)₂ (blue) in complex with IST1₃₂₂₋₃₆₆ (green, with buried core interface sidechains shown) (PDB 8UC6). Locations of residues that were mutated in CAPN7(MIT)₂ are shown in magenta, turquoise, and red. Dashed lines show CAPN7 and IST1 residues that lack well-defined electron density (see 'Materials and methods'). (**B**) Structure of the CAPN7₄₋₆₈–IST1₃₂₄₋₃₃₂ complex (blue and green) overlaid with the structure of VPS4A₃₋₇₅-CHMP6₁₆₈₋₁₇₉ MIM complex (gray and light blue, PDB 2K3W). Note the similar

*Figure 2 continued on next page*

*Figure 2 continued*

binding type 2 MIT-MIM binding modes. (**C**) Structure of the CAPN7$_{80-150}$-IST1$_{352-363}$ complex (blue and green) overlaid with the structure of SPARTIN$_{8-101}$-IST1$_{352-363}$ complex (gray and light blue, PDB 4U7I). Note the similar type 3 MIT-MIM binding modes.

The online version of this article includes the following figure supplement(s) for figure 2:

**Figure supplement 1.** Unbiased electron density omit map for IST1$_{322-366}$.

**Figure supplement 2.** Alignment of IST1$_{325-336}$ and CHMP6$_{168-179}$ in the CAPN7$_{4-68}$ binding groove.

**Figure supplement 3.** Binding isotherms for individual CAPN7 microtubule-interacting and trafficking (MIT) domains binding to individual IST1 MIT-interacting motif (MIM) elements.

**Table 1.** CAPN7(MIT)$_2$–IST1$_{322-366}$ complex (PDB: 8UC6) crystallographic data and refinement statistics.

| Data collection, integration, and scaling | |
|---|---|
| Programs | XDS, AIMLESS |
| Source/wavelength (Å) | SSRL 14–1/1.19499 |
| Space group (unit cell dimensions) | P6$_5$22 (87.84, 87.84, 183.89, 90.0, 90.0, 120.0) |
| Resolution (high-resolution shell) (Å) | 40.0–2.70 (2.83–2.70) |
| # reflections measured | 1,398,023 |
| # unique reflections | 12,228 |
| Redundancy (high-resolution shell) | 114 (104) |
| Completeness (high-resolution shell) (%) | 100.0 (99.9) |
| <I/σI> (high-resolution shell) | 11.0 (1.5) |
| <CC1/2> | 0.998 (0.650) |
| Rpim (high-resolution shell) | 0.080 (0.666) |
| Mosaicity (°) | 0.12 |

| Refinement | |
|---|---|
| Program | Phenix.refine |
| Resolution (Å) | 40.0–2.70 |
| Resolution (Å) – (high-resolution shell) | (2.81–2.70) |
| # reflections | 12,169 |
| # reflections in Rfree set excluded from refinement | 1221 |
| Rcryst | 0.211 (0.284) |
| Rfree | 0.285 (0.371) |
| RMSD: bonds (Å)/angles (°) | 0.008/0.976 |
| B-factor refinement | Group B |
| <B> (Å$^2$): all atoms/# atoms | 49/2,779 |
| <B> (Å$^2$): water molecules/#water | 46/42 |
| Φ/φ most favored (%)/additionally allowed (%) | 97/1.8 (0.9 outlier) |

CC1/2 = correlation coefficient. Rpim = precision-indicating merging R-factor. RMSD = root-mean-square deviation.

The C-terminal IST1 MIM element ($_{349}$DIDFDDLSRRFEELKK$_{364}$) forms an amphipathic helix that packs between helices 1 and 3 of the C-terminal CAPN7 MIT domain, making a canonical 'type 3' (MIM3) interaction (*Figure 2A*, bottom, and *Figure 2—figure supplement 1B*; *Yang et al., 2008*; *Skalicky et al., 2012*; *Wenzel et al., 2022*). Core residues from the hydrophobic face of the IST1 helix are buried in the CAPN7 MIT helix 1/3 groove, and the interface hydrophobic contacts, hydrogen bonds, and salt bridges are nearly identical to those seen when this same IST1 motif makes a type 3 interaction with the MIT domain of SPARTIN (*Figure 2C*; *Guo and Xu, 2015*; *Wenzel et al., 2022*). A previous study has reported that Thr95 of CAPN7 can be phosphorylated (*Mayya et al., 2009*). This residue sits in the IST1$_{349-364}$ binding site, adjacent to the detrimental F98D mutation (see below), and Thr95 phosphorylation would position the phosphate near Leu355 of IST1, creating an unfavorable electrostatic interaction and steric clash. We therefore anticipate that Thr95 phosphorylation would reduce IST1 binding and could negatively regulate the CAPN7-IST1 interaction.

The selectivity of each CAPN7 MIT domain for its cognate IST1 MIM element binding is dictated by the character of the MIT binding grooves. The hydrophobic contacts and hydrogen bonding potential of the residues within the N-terminal MIT domain are selective for type 2 interactions (IST1$_{324-332}$), whereas the C-terminal MIT domain selects for type 3 interactions. This is consistent with our binding data for each individual CAPN7 MIT with each individual IST1 MIM element (*Figure 2—figure supplement 3*).

In summary, the CAPN7(MIT)$_2$-IST1$_{322-366}$ structure (1) provides the first example of IST1$_{325-336}$ bound to an MIT domain and confirms the expectation that this IST1 element engages in MIM2-type interactions, (2) demonstrates that IST1$_{352-363}$ engages in a MIM3-type interaction when it binds CAPN7, and (3) shows that the separate IST1 MIM elements each bind distinct MIT domains within CAPN7(MIT)$_2$.

## Mutational analyses of the CAPN7–IST1 complex

In our earlier study, we identified point mutations in the helix 1/3 grooves of each MIT domain that disrupt IST1 binding (*Wenzel et al., 2022*). The crystal structure of the CAPN7(MIT)$_2$–IST1$_{322-366}$ complex explains why IST1 binding is disrupted by the V18D mutation in the first CAPN7 MIT domain (magenta, *Figure 2A*, top) and by the F98D mutation in the second MIT domain (red, *Figure 2A*, bottom), as replacing either of these hydrophobic residues with charged residues would create unfavorable interactions in the hydrophobic cores of the MIT binding interfaces. We also used the structure to design a control helix 1/2 mutation that was not expected to alter IST1 binding (L61D, cyan, *Figure 2A*, top). Fluorescence polarization anisotropy binding assays showed that the single V18D or F98D mutations in CAPN7(MIT)$_2$ each reduced IST1$_{316-366}$ binding >20-fold, with a greater effect seen for the N-terminal MIT mutation (V18D) (*Figure 3A*). The V18D/F98D double mutation decreased IST1$_{316-366}$ binding even further (>300-fold vs. wt CAPN7(MIT)$_2$). As predicted, the control L61D mutation did not affect IST1$_{316-366}$ binding. Importantly, none of these mutations significantly disrupted the overall CAPN7(MIT)$_2$ protein fold as assessed by circular dichroism spectroscopy (*Figure 3—figure supplement 1A*). Thus, we have identified CAPN7(MIT)$_2$ mutations that diminish IST1$_{316-366}$ complex formation by specifically disrupting each of the two MIT-MIM binding interfaces.

We also assessed the importance of these interfaces for association of the full-length IST1 and CAPN7 proteins, using size-exclusion chromatography (SEC) to analyze the individual proteins and their complex (*Figure 3B* and *Figure 3—figure supplement 2*). SEC with multi-angle light scattering (SEC-MALS) analyses showed that the individual IST1 and CAPN7 proteins both eluted as monomers (IST1: calculated MW = 40.0 kDa, SEC-MALS estimated MW = 41 ± 1 kDa; CAPN7 calculated MW = 92.6 kDa, SEC-MALS estimated MW = 90.4 ± 0.3 kDa, see *Figure 3—figure supplement 2B*). However, IST1 eluted anomalously rapidly, apparently because it adopts an extended conformation. When mixed, full-length IST1 and CAPN7 formed a 1:1 complex that migrated even more rapidly. Importantly, complex formation was inhibited by inactivating mutations in the center of the two IST1 MIM elements interaction interfaces (L328D/L355A, see *Figure 2A* and *Figure 3—figure supplement 3*; *Obita et al., 2007*; *Stuchell-Brereton et al., 2007*; *Kieffer et al., 2008*; *Bajorek et al., 2009*). Thus, the crystallographically defined MIT-MIM interactions also mediate association of the full-length IST1 and CAPN7 proteins in solution. Finally, we analyzed the CAPN7-IST1 interaction in a cellular context using co-immunoprecipitation assays that employed our panel of different CAPN7 MIT mutations. Epitope-tagged CAPN7 and IST1 constructs were co-expressed in HEK293T cells, CAPN7-OSF constructs were immunoprecipitated from cellular extracts, with co-precipitated

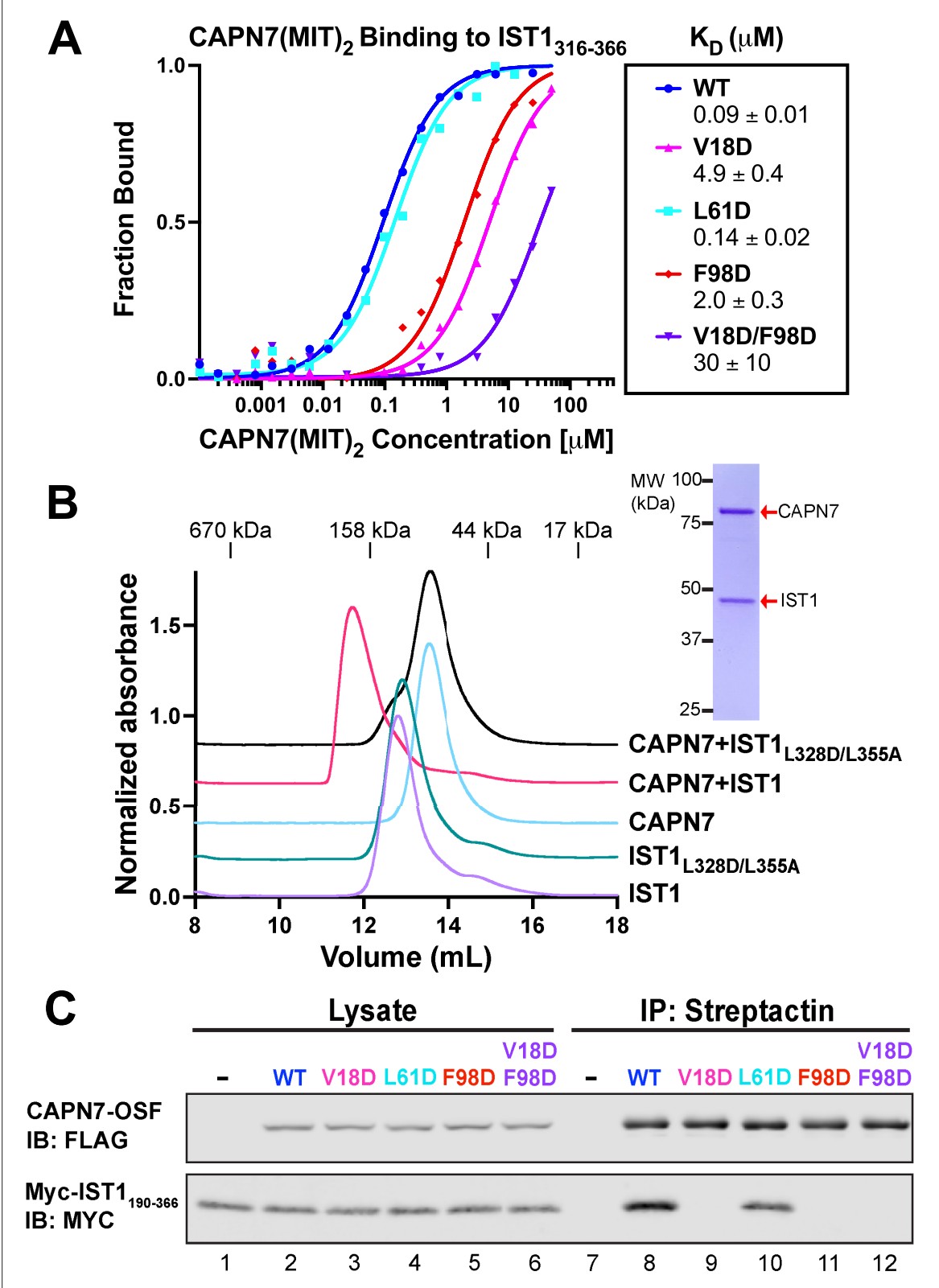

**Figure 3.** Mutational analyses of the CAPN7-IST1 complex. (**A**) Fluorescence polarization anisotropy binding isotherms showing CAPN7(MIT)$_2$ constructs binding to IST1$_{316-366}$. Isotherm data points and dissociation constants are means from three independent experiments ± standard error of the mean. WT, V18D, and F98D binding isotherms are reproduced from *Wenzel et al., 2022* for comparison. (**B**) Size-exclusion chromatography binding analyses of free proteins or 1:1 mixtures of full-length CAPN7 with WT (black) or L328D/L355A mutant (green) full-length IST1. Note that the IST1 mutations disrupt

*Figure 3 continued on next page*

*Figure 3 continued*

CAPN7-IST1 complex formation. Inset image shows a Coomassie-stained SDS-PAGE gel of the peak fraction from the CAPN7+ IST1 chromatogram. (**C**) Co-immunoprecipitation of Myc-IST1$_{190-366}$ with the indicated full-length CAPN7-OSF constructs from extracts of transfected HEK293T cells.

The online version of this article includes the following source data and figure supplement(s) for figure 3:

**Source data 1.** Annotated and uncropped Coomassie-stained SDS-PAGE for *Figure 3B*.

**Source data 2.** Annotated and uncropped western blots and raw images for *Figure 3C*.

**Figure supplement 1.** Circular dichroism spectra of recombinant CAPN7(MIT)$_2$ proteins and co-immunoprecipitation of full-length CAPN7 and full-length IST1 from cells.

**Figure supplement 1—source data 1.** Annotated and uncropped western blots and raw images for *Figure 3—figure supplement 1B*.

**Figure supplement 2.** Size-exclusion chromatographic analyses of CAPN7 and IST1 complex formation.

**Figure supplement 3.** CAPN7(MIT)$_2$ binding to IST1$_{316-366}$ is diminished by mutations in either IST1 MIT-interacting motif (MIM) element.

Myc-IST$_{190-366}$ (*Figure 3*) or Myc-IST1 (*Figure 3—figure supplement 1B*) detected by western blotting. Wild-type CAPN7 and the control CAPN7(L61D) mutant both co-precipitated IST1$_{190-366}$ and IST1 efficiently, whereas all three inactivating CAPN7 MIT point mutations (V18D, F98D, and V18D/F98D) dramatically reduced the co-precipitation of Myc-IST$_{190-366}$ (no detectable binding) or Myc-IST1 (very low-level residual binding, indicating either non-specific background or minor contributions from other region(s) of the protein). Thus, both crystallographically defined MIT:MIM binding interfaces mediate association of full-length CAPN7 and IST1 in vitro and in cell extracts.

## IST1 recruits CAPN7 to the midbody

We next tested the midbody localization of wild-type CAPN7 and CAPN7 mutants lacking IST1 binding (V18D, F98D) or proteolytic (C290S) activities (*Osako et al., 2010*). These studies employed HeLa cell lines treated with siRNAs to deplete endogenous CAPN7, and concomitantly induced to express integrated, siRNA-resistant, mCherry-tagged CAPN7 rescue constructs. Nup153 depletion was also used to maintain NoCut checkpoint signaling (*Mackay et al., 2010*; *Strohacker et al., 2021*; *Wenzel et al., 2022*), and thymidine treatment/washout was used to synchronize cell cycles and increase the proportion of midbody-stage cells (*Figure 4—figure supplement 1*). As expected, IST1 localized in a double-ring pattern on either side of the central Flemming body within the midbody (*Agromayor et al., 2009*; *Bajorek et al., 2009*), and wild-type CAPN7-mCherry colocalized with IST1 in the same pattern (*Wenzel et al., 2022*). CAPN7(C290S)-mCherry similarly colocalized with endogenous IST1 on either side of the Flemming body, whereas the two IST1 binding mutants did not (*Figure 4A*). IST1 midbody localization was normal in all cases. Quantification revealed that wild-type CAPN7 and CAPN7(C290S) colocalized with IST1 in ~80% of all IST1-positive midbodies, whereas the two IST1 non-binding mutants colocalized with IST1 in <5% of midbodies (*Figure 4B*; *Wenzel et al., 2022*). These data are consistent with our previous report that the V18D IST1-binding mutation did not localize to midbodies (*Wenzel et al., 2022*). We conclude that IST1 recruits CAPN7 to midbodies, and efficient localization requires that both CAPN7 MIT domains bind both IST1 MIM elements, whereas CAPN7 proteolytic activity is not required for proper localization.

## IST1-binding and proteolytic activity are required for CAPN7 functions in abscission and NoCut checkpoint maintenance

Our previous study indicated that CAPN7 promotes both abscission and NoCut checkpoint regulation (*Wenzel et al., 2022*). Here, we tested the requirements for CAPN7 to function in each of these processes using increased frequencies of cells with midbody connections and multiple nuclei as surrogate markers for abscission defects and NoCut-induced abscission delays. As expected, depletion of CAPN7 from asynchronous HeLa cells increased the fraction of cells that stalled or failed at abscission, as reflected by significant increases in midbody stage cells (from 5 to 10%) and multinucleate cells, indicating failed abscission (from 6 to 22%) vs. control treatments (*Figure 5A and B*, *Figure 5—figure supplement 1A*). Efficient abscission was almost completely rescued by expression of wild-type CAPN7-mCherry, whereas abscission defects were not rescued by CAPN7 constructs that were deficient in IST1-binding (V18D or F98D) or proteolysis (C290S). Thus, both IST1-binding and proteolytic activity are required for CAPN7 to support efficient abscission.

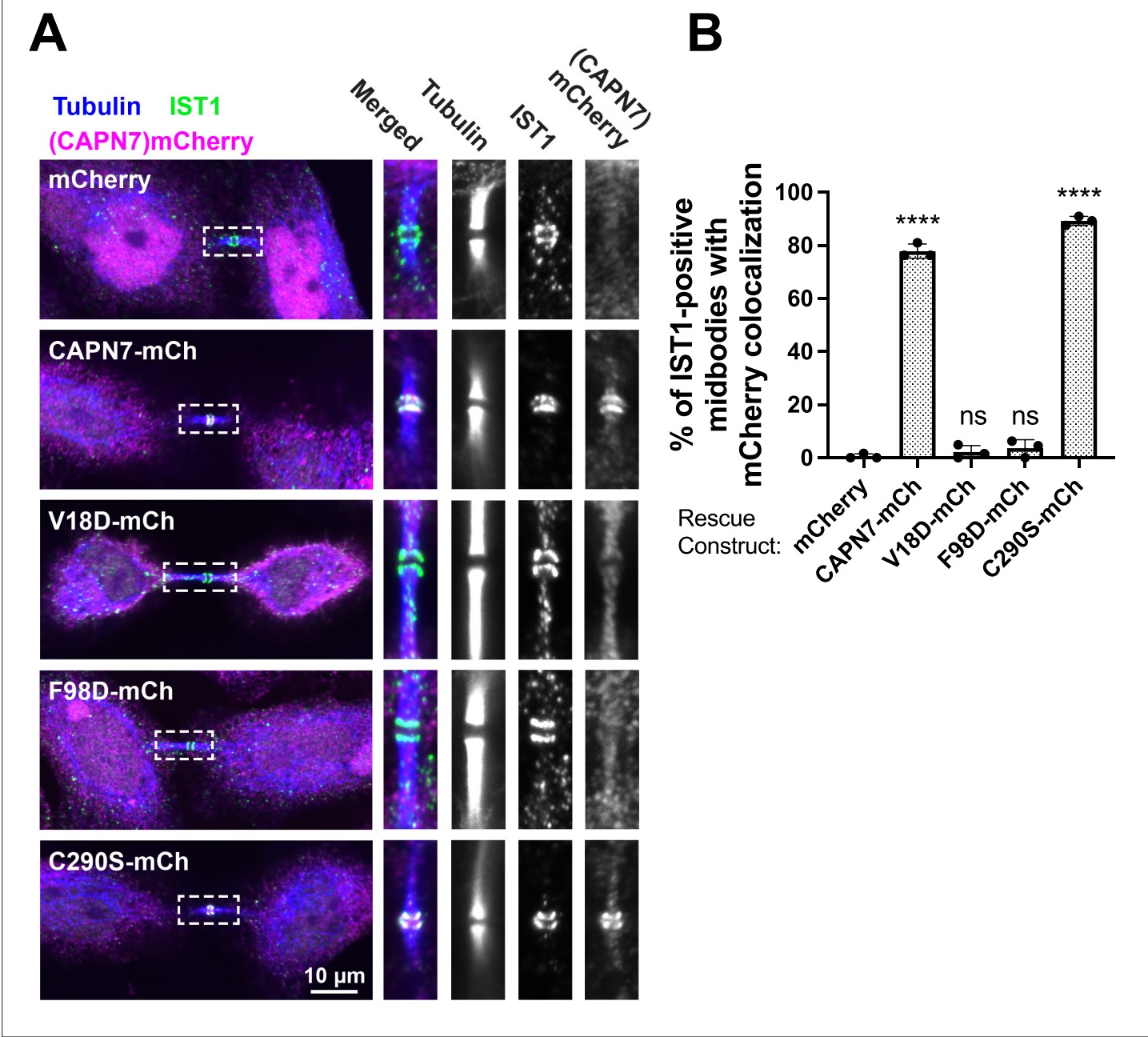

**Figure 4.** IST1 binding is required for CAPN7 midbody localization. (**A**) Representative immunofluorescence images showing the extent of midbody colocalization of mCherry-CAPN7 constructs (or the mCherry control), with endogenous IST1 in synchronous, NoCut checkpoint-active cells. Endogenous CAPN7 was depleted with siRNA while siRNA-resistant CAPN7-mCherry constructs were inducibly expressed. (**B**) Quantification of the colocalization of mCherry-CAPN7 constructs with endogenous IST1 at midbodies (corresponding to the images in **A**). Colocalization was scored blinded as described in 'Materials and methods.' Bars represent the mean and standard error of the mean from three independent experiments where > 50 IST1-positive midbody-stage cells were counted per experiment. Statistical analyses were performed using unpaired *t*-tests that compared the percentage of rescue constructs that colocalized with IST1 at midbodies to the mCherry alone control. ****$p < 0.0001$, ns (not significant) $p > 0.05$.

The online version of this article includes the following source data and figure supplement(s) for figure 4:

**Figure supplement 1.** Western blot confirmation of rescue construct expression and siRNA knockdown efficiency.

**Figure supplement 1—source data 1.** Annotated and uncropped western blots and raw images for *Figure 4—figure supplement 1*.

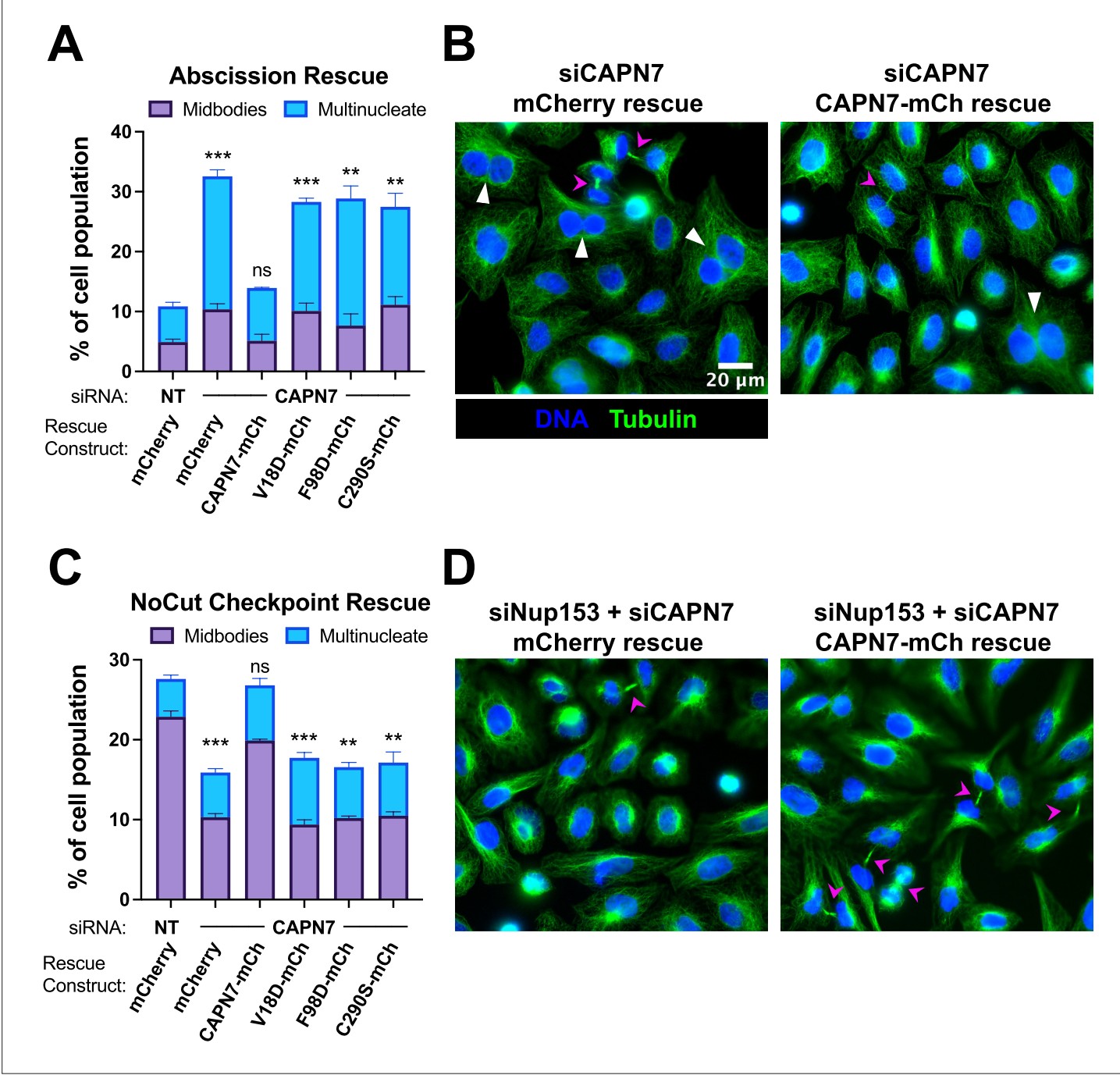

**Figure 5.** IST1-binding and catalytic activity are required for CAPN7 abscission and NoCut functions. (**A, C**) Quantification of midbody-stage and multinucleate HeLa cells from unperturbed asynchronous cultures (**A**), or cells in which NoCut checkpoint activity was sustained by Nup153 depletion (**C**). (**B, D**) Representative images from the quantified datasets in (**A and C**), respectively. Midbodies are marked with magenta arrows, and multinucleate cells are marked with white arrowheads. In all cases, cells were depleted of endogenous CAPN7, followed by expression of the designated DOX-inducible 'rescue' construct. Bars represent the mean and standard error of the mean from three independent experiments where >300 cells were counted per experiment. Statistical analyses were performed using unpaired *t*-tests comparing the sum of midbody-stage and multinucleate cells for each individual treatment to the same sum in siNT (non-targeting) control-treated cells. ***p<0.001, **p<0.01, ns (not significant) p>0.05.

The online version of this article includes the following source data and figure supplement(s) for figure 5:

**Figure supplement 1.** Western blot confirmation of rescue construct expression and siRNA knockdown efficiency.

**Figure supplement 1—source data 1.** Annotated and uncropped western blots and raw images for *Figure 5—figure supplement 1*.

We also tested the requirements for CAPN7 to support NoCut checkpoint signaling. In these experiments, NoCut signaling was maintained by Nup153 depletion, which delays abscission and raises the proportion of midbody-stage cells to 23% (*Figure 5C and D*, *Figure 5—figure supplement 1B*; *Mackay et al., 2010*). Co-depletion of CAPN7 reduced the percentage of cells stalled at the midbody stage to 10%, indicating that CAPN7 is required to maintain the NoCut checkpoint signaling induced by Nup153 knockdown, in good agreement with our previous study (*Wenzel et al., 2022*). Rescue with wild-type CAPN7-mCherry restored NoCut checkpoint function almost completely (20% midbodies), whereas the IST1-binding or catalytically dead mutant CAPN7 constructs failed to rescue the checkpoint (all ~10% midbodies). Hence, CAPN7 must bind IST1 and be an active protease to participate in NoCut checkpoint regulation.

## CAPN7 and SPAST are required to sustain NoCut signaling in response to DNA bridges and replication stress

CAPN7 and the AAA ATPase Spastin (SPAST) are required to sustain NoCut signaling when the checkpoint is triggered by co-depletion of nuclear pore proteins Nup153 and Nup50 (*Wenzel et al., 2022*). Here, we tested whether CAPN7 and SPAST are also required for NoCut when the checkpoint is triggered in other ways, including the presence of mis-segregated DNA in the midbody (here termed DNA bridges) (*Steigemann et al., 2009*; *Mendoza et al., 2009*; *Bembenek et al., 2013*) or replication stress (*Mackay and Ullman, 2015*). DNA bridges were induced using the topoisomerase II inhibitor, ICRF-193, which inhibits DNA untangling and thereby promotes mis-segregation (*Clarke et al., 1993*; *Germann et al., 2014*; *Nielsen et al., 2015*; *Bhowmick et al., 2019*; *Jiang et al., 2023*). Treatment of asynchronous control HeLa cells with a low dose of ICRF-193 (80 nM) induced NoCut signaling, as reflected by increases in the proportion of cells with midbodies (from 5 to 11%) and multiple nuclei (from 4 to 8%) (*Figure 6A*), in good agreement with previous reports (*Bhowmick et al., 2019*). As expected, control depletion of Katanin p60 (KATNA1), a microtubule severing AAA ATPase required for abscission but not NoCut signaling (*Matsuo et al., 2013*; *Wenzel et al., 2022*), impaired abscission, as reflected by increases in the frequencies of midbodies (from 5 to 9%) and multinucleate cells (from 4 to 11%). ICRF-193 treatment further increased these values (with midbodies increasing from 9 to 16% and multinucleate cells increasing from 11 to 16%), reflecting the additive effects of impaired abscission and NoCut signaling. Cells depleted of either SPAST or CAPN7 also showed the expected abscission impairments, which were particularly dramatic for CAPN7, where midbodies increased from 5 to 9% and multinucleate cells increased from 4 to 20%. In these cases, however, ICRF-193 treatments did not induce further abscission defects/delays, implying that NoCut was not active in the absence of these proteins and therefore that SPAST and CAPN7 are required to sustain NoCut checkpoint signaling induced by DNA bridges.

We also tested the requirement for SPAST and CAPN7 to support the NoCut checkpoint in response to replication stress induced by the DNA polymerase inhibitor aphidicolin (*Sheaff et al., 1991*; *Chan et al., 2009*; *Lukas et al., 2011*; *Harrigan et al., 2011*; *Mackay and Ullman, 2015*; *Sadler et al., 2018*). As shown in *Figure 6B*, the results were essentially identical to those obtained when NoCut signaling was induced by ICRF-193 treatments. Specifically, control treatments of asynchronous HeLa cells with low doses of aphidicolin (0.4 µM) maintained checkpoint activation in both the presence and absence of KATNA1. In contrast, cells depleted of SPAST or CAPN7 did not exhibit NoCut checkpoint responses following aphidicolin treatments, indicating that both proteins are required to sustain NoCut signaling when the checkpoint is triggered by replication stress.

Our discovery that both CAPN7 and SPAST participate in the competing processes of cytokinetic abscission and NoCut delay of abscission may appear counterintuitive, but we envision that the MIT proteins could participate in both processes if they change substrate specificities or activities when participating in NoCut vs. abscission; for example, via different sites of action, post-translational modifications, and/or binding partners. We note that, in addition to its well-documented function in clearing spindle microtubules to allow efficient abscission (*Yang et al., 2008*), SPAST is also required for ESCRT-dependent closure of the nuclear envelope (NE) (*Vietri et al., 2015*). The relationship between NE closure and NoCut signaling is not yet well understood, and it is therefore conceivable that nuclear membrane integrity is required to allow mitotic errors to sustain NoCut signaling. It will therefore be of interest to determine whether or not CAPN7, in addition to its midbody abscission

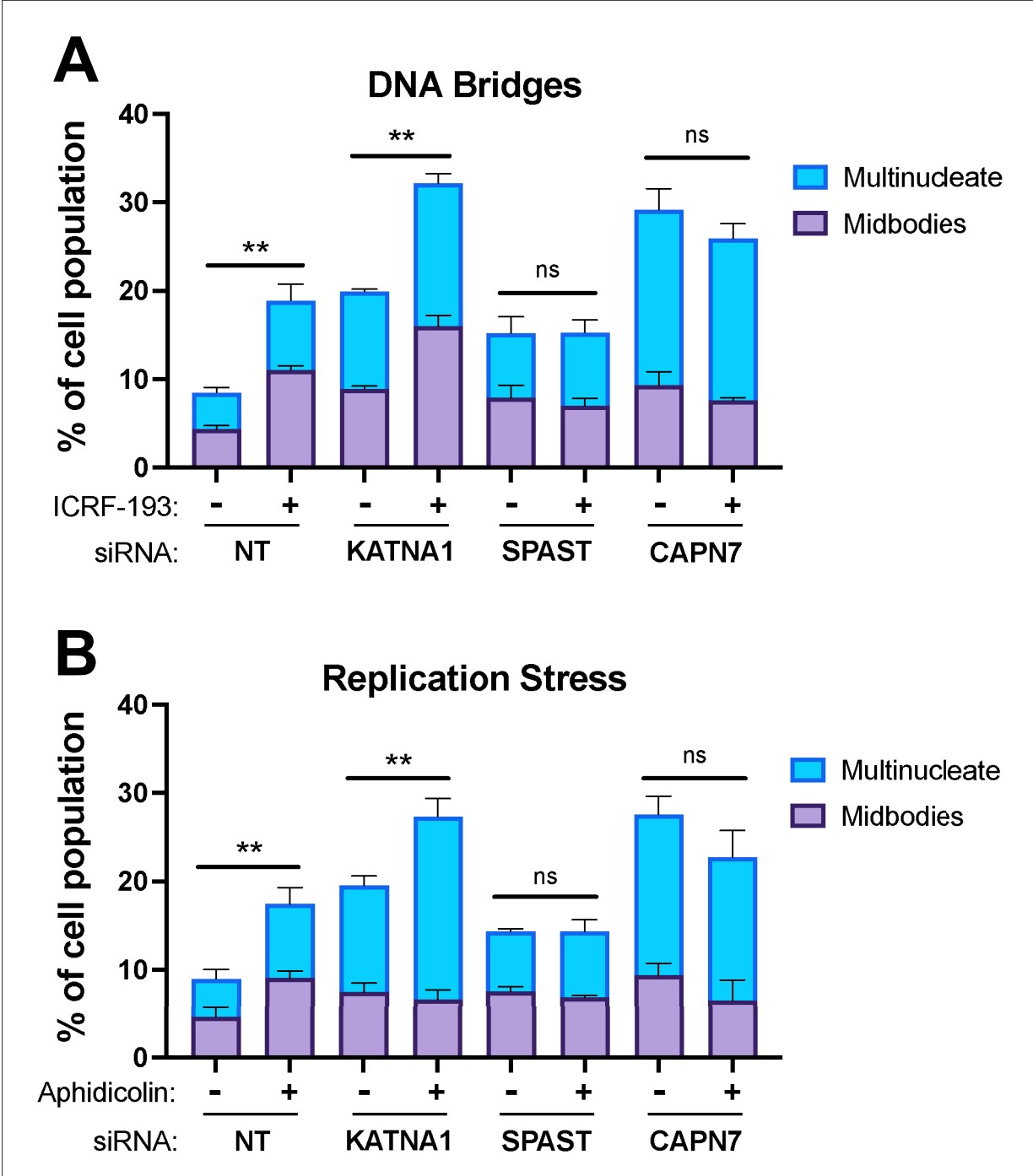

**Figure 6.** CAPN7 and SPAST are required to maintain the NoCut checkpoint in response to DNA bridges and replication stress. Quantification of midbody-stage and multinucleate HeLa cells from asynchronous cultures in which NoCut checkpoint activity was sustained by inducing DNA bridges with ICRF-193 treatment (**A**) or replication stress with aphidicolin treatment (**B**), following siRNA treatments to deplete the indicated proteins. Bars represent the mean and standard error of the mean from three independent experiments where >300 cells were counted per experiment. Statistical analyses were performed using unpaired *t*-tests comparing the sum of midbody-stage and multinucleate cells for each siRNA treatment with or without NoCut checkpoint maintenance with the indicated drug. **p<0.01, ns (not significant) p>0.05.

functions, also participates in nuclear envelope closure and, if so, whether that activity is connected to its NoCut functions.

## CAPN7 as an evolutionarily conserved ESCRT-signaling protein

Our data imply that proteolysis is critical for CAPN7 functions in NoCut and abscission, although physiological substrates have not yet been identified. GFP-CAPN7 can perform autolysis, and this activity is enhanced by IST1 binding (*Osako et al., 2010*), implying that interactions with IST1 could activate CAPN7 proteolytic activity. In addition to our studies implicating CAPN7 midbody functions during cytokinesis, other biological functions for CAPN7 have also been proposed, including in ESCRT-dependent degradation of EGFR through the endosomal/MVB pathway (*Yorikawa et al., 2008*; *Maemoto et al., 2014*), and in human endometrial stromal cell migration and invasion (*Liu et al., 2013*) and decidualization (*Kang et al., 2022*). Aberrant CAPN7 expression may also impair embryo implantation through a currently unknown mechanism (*Yan et al., 2018*).

CAPN7 has evolutionarily conserved orthologs throughout Eukaryota, including in Aspergillus and yeast, where its role in ESCRT-dependent pH signaling is well described (*Denison et al., 1995*; *Orejas et al., 1995*; *Futai et al., 1999*; *Peñalva et al., 2014*). In Aspergillus, pH signaling requires proteolytic activation of the transcription factor PacC (no known human ortholog), which initiates an adaptive transcriptional response. PacC is cleaved and activated by the MIT domain-containing, Calpain-like CAPN7 ortholog protease PalB, which is recruited to membrane-associated ESCRT-III filaments in response to alkaline pH through a MIT:MIM interaction with the ESCRT-III protein Vps24 (CHMP3 ortholog) (*Rodríguez-Galán et al., 2009*). These ESCRT assemblies also contain PalA (ALIX ortholog), which co-recruits PacC to serve as a PalB substrate (*Xu and Mitchell, 2001*; *Vincent et al., 2003*).

The pH signaling system in fungi has striking parallels with our model for human CAPN7 functions in cytokinesis, including: (1) MIT:MIM interactions between CAPN7 and ESCRT-III, which serve to recruit the protease to a key ESCRT signaling site. (2) The local presence of the homologous PalA and ALIX proteins. In the pH sensing system, PalA is thought to act as a scaffold that 'presents' the substrate to PalB for proteolysis on ESCRT-III assemblies, thus providing an extra level of spatiotemporal specificity for proteolysis. In the mammalian midbody, ALIX helps to nucleate ESCRT-III filament formation. (3) A requirement for proteolysis in downstream signal transduction. Thus, ESCRT-III assemblies are evolutionarily conserved platforms that recruit and unite active calpain proteases and their substrates with spatiotemporal specificity to regulate key biological processes.

# Materials and methods

**Key resources table**

| Reagent type (species) or resource | Designation | Source or reference | Identifiers | Additional information |
|---|---|---|---|---|
| Cell line (*Homo sapiens*) | Hela-N | Maureen Powers Lab | | HeLa cells selected for transfectability |
| Cell line (*Homo sapiens*) | HEK293T | ATCC | CRL-3216 | |
| Antibody | Anti-FLAG (M2, mouse monoclonal) | Sigma | F1804 | WB (1:5000) |
| Antibody | Anti-MYC (4A6, mouse monoclonal) | Millipore | 05-724 | WB (1:2500) |
| Antibody | Anti-RFP (rat monoclonal) | ChromoTek | 5F8 | IF (1:500) |
| Antibody | Anti-RFP (mouse monoclonal) | ChromoTek | 6G6 | WB (1:1000) |
| Antibody | Anti-alpha-tubulin (DM1A, mouse monoclonal) | Cell Signaling Technologies | DM1A | IF (1:2000) |
| Antibody | Anti-alpha-Tubulin (chicken polyclonal) | Synaptic Systems | 302 206 | IF (1:1000) |
| Antibody | Anti-CAPN7 (rabbit polyclonal) | Proteintech | Cat# 26985-1-AP | IF (1:500) WB (1:4000) |
| Antibody | Anti-IST1 (rabbit polyclonal) | Sundquist Lab/Covance | UT560 | IF (1:1000) |

*Continued on next page*

*Continued*

| Reagent type (species) or resource | Designation | Source or reference | Identifiers | Additional information |
|---|---|---|---|---|
| Antibody | Anti-NUP153 (SA1) (mouse monoclonal) | Brian Burke | | WB (1:50) |
| Antibody | Anti-NUP50 (rabbit polyclonal) | *Mackay et al., 2010* | | WB (1:2500) |
| Antibody | Anti-GAPDH (mouse monoclonal) | Millipore | | |
| Sequence-based reagent | siNT | *Mackay et al., 2010* | siRNA | GCAAAUCUCCGAUCGUAGA |
| Sequence-based reagent | siCAPN7 | *Wenzel et al., 2022* | siRNA | GCACCCAUACCUUUACAUU |
| Sequence-based reagent | siNUP153 | *Mackay et al., 2010* | siRNA | GGACUUGUUAGAUCUAGUU |
| Chemical compound, drug | Doxycycline Hyclate | Sigma | 324385 | 1–2 µg/mL |
| Chemical compound, drug | Thymidine | CalBiochem | CAS 50-89-5 | 2 mM |
| Chemical compound, drug | Oregon Green 488 maleimide | Life Technologies/Molecular Probes | O6034 | Fluorescent label for peptides |
| Software, algorithm | Fiji | NIH | RRID:SCR_002285 | |
| Software, algorithm | Prism 9 | GraphPad | | |

## Cloning

All plasmids created for this study were made by Gibson Assembly (*Gibson et al., 2009*) using vectors linearized by restriction enzymes (New England Biolabs). pCA528 bacterial expression plasmids expressing His$_6$-SUMO-tagged fusion proteins with a native N-terminus after tag cleavage were linearized with BsaI and BamHI. pCAG-OSF mammalian expression plasmids that encoded proteins for co-IP binding assays were linearized with KpnI and XhoI. pLVX vectors used for generating inducible cell lines were linearized by BamHI and MluI. Point mutations were generated using QuikChange II Site-directed mutagenesis kit (Agilent). See *Supplementary file 2* for complete details of the plasmids used in our studies.

## Bacterial expression of (His)$_6$-SUMO-fusion proteins

Proteins were expressed in BL21 RIPL cells grown in ZYP-5052 autoinduction media (*Studier, 2005*). Transformed cells were initially grown for 3–6 hr at 37°C to an OD600 of 0-4-0.6, and then switched to 19°C for an additional 20 hr. Cells were harvested by centrifugation at 6000 × *g*, and cell pellets were stored at –80°C.

## Purification of (His)$_6$-SUMO-fusion proteins

All purification steps were carried out at 4°C except where noted. Frozen cell pellets were thawed and resuspended in lysis buffer: 50 mM Tris (pH 8.0 at 23°C), 500 mM NaCl, 2 mM imidazole, 1 mM dithiothreitol (DTT), supplemented with 0.125% sodium deoxycholate, lysozyme (25 µg/mL), PMSF (100 µM), pepstatin (10 µM), leupeptin (100 µM), aprotinin (1 µM), DNAseI (10 µg/mL), and 2 mM MgCl$_2$. Cells were lysed by sonication and lysates were clarified by centrifugation at 40,000 × *g* for 45 min. Clarified supernatant was filtered through a 0.45 µM PES syringe filter and incubated with 10 mL of cOmplete His-Tag purification beads (Roche) for 1 hr with gentle rocking. Beads were washed with 500 mL unsupplemented lysis buffer. Fusion proteins were eluted with 50 mL of lysis buffer supplemented with 250 mM imidazole. Eluted proteins were treated with 100 µg His$_6$-ULP1 protease overnight in 6–8k MWCO dialysis bags while dialyzing against 2 × 2 L of 25 mM Tris pH (8.0 at 23°C), 100 mM NaCl, 1 mM TCEP, 0.5 mM EDTA. The dialysate was purified by Q Sepharose chromatography (HiTrap Q HP 5 mL; Cytiva Life Sciences) with a linear gradient elution from 100 to 1000 mM NaCl. Fractions containing processed fusion proteins were then passed over 5 mL of cOmplete His-Tag purification beads to remove residual uncleaved His$_6$-Sumo-fusion protein and His$_6$-Sumo cleaved tag. Proteins were concentrated and purified by Superdex 75 or Superdex 200 gel filtration chromatography (120 mL; 16/600; Cytiva Life Sciences) in 25 mM Tris (pH 7.2 at 23°C), 150 mM NaCl, 1 mM TCEP, and 0.5 mM EDTA. Highly pure fractions were pooled and concentrated. Masses of each protein were confirmed using ESI-MS (University of Utah Mass Spectrometry Core Facility, see *Supplementary*

*file 1*). Yields ranged from 0.2 to 50 mg/L of bacterial culture. Yields of full-length CAPN7 were low (~0.2 mg/L), which is why full-length IST1 and mutants (~20 mg/L yields) were used for gel filtration binding experiments in *Figure 3B*.

## Peptide fluorescent labeling

Fluorescent labeling was performed by the University of Utah DNA/Peptide Synthesis Core as described previously (*Caballe et al., 2015*; *Talledge et al., 2018*). Briefly, peptides were labeled in DMSO using ~1.3-fold molar excess of Oregon Green 488 maleimide (Life Technologies/Molecular Probes #O6034) dissolved in a 1:1 solution of acetonitrile:DMSO. Reversed-phase HPLC was used to monitor the reactions and separate labeled peptides from unreacted dye and unlabeled peptides. Labeled peptide fractions were dried under vacuum and dissolved in water. Peptide concentrations were quantified using the absorbance of Oregon Green 488 at 491 nm ($\varepsilon = 83{,}000$ cm$^{-1}$ M$^{-1}$ in 50 mM potassium phosphate, pH 9.0).

## Fluorescence polarization anisotropy-binding assays

FPA-binding experiments were performed as described previously (*Caballe et al., 2015*; *Wenzel et al., 2022*) in 25 mM Tris, pH 7.2, 150 mM NaCl, 0.1 mg/mL bovine serum albumin (BSA), 0.01% Tween-20, and 0.5 mM TCEP, with 250 pM fluor-labeled IST1 peptides and purified CAPN7 MIT domain constructs at the indicated concentrations. A Biotek Synergy Neo Multi-Mode plate reader was used to measure fluorescence polarization with excitation at 485 nm and emission (detection) at 535 nm. Binding isotherms were fit to 1:1 models using Prism 9 (GraphPad). Raw data were first fit to the equation $Y = B_{max} * X/(K_d + X) + m2$, each binding replicate isotherm was then normalized to a fraction bound value by subtracting the fitted m2 baseline value and dividing by the fitted Bmax value. Following normalization, three replicate isotherms were merged to generate an average $K_D$, treating each of the replicate values as individual data points. Reported $K_D$ values indicate the average ± standard error of the mean from three independent binding experiments. In cases where binding isotherms did not reach half occupancy at the highest ligand concentration tested, we simply report $K_D$ as being greater than the highest concentration used (as in *Figure 1B*, IST1$_{316-343}$). The $K_D$ are listed as not determined (ND) if changes in fluorescence polarization anisotropy were too small to allow meaningful curve fitting (as in *Figure 2—figure supplement 3*).

## NMR spectroscopy

NMR data were collected on a Varian INOVA 600 MHz spectrometer running OpenVnmrJ_v3.1. 2D $^{15}$N-HSQC data were processed using nmrPipe (*Delaglio et al., 1995*) and spectra were analyzed and NMR figures prepared using NMRFAM-SPARKY (*Lee et al., 2015*). NMR resonance assignments for $^{15}$N-labeled IST1$_{303-366}$ were taken from *Caballe et al., 2015*. Briefly, sequential assignments were obtained from 2D $^{15}$N-HSQC and 3D HNCA, HNCACB, CBCAcoNH, HNCO, and HNcaCO triple resonance experiments. Using this approach, it was possible to assign all 55 non-proline backbone amide resonances. Chemical shift assignments were deposited in the BMRB (accession no: 25393).

Chemical shifts and peak intensity changes induced by CAPN7(MIT)$_2$ binding to IST1$_{303-366}$ were measured by titrating 0.25 mM of $^{15}$N-labeled IST1$_{303-366}$ with increasing amounts of unlabeled CAPN7(MIT)$_2$ to final stoichiometries of IST1:CAPN7(MIT)$_2$ of 1:0, 1:0.25, 1:0.5, 1:1, and 1:1.3. Both reagents were in NMR buffer containing 20 mM sodium phosphate (pH 6.8), 5% D$_2$O, 100 mM NaCl, 0.1 mM TCEP, and 0.1 mM EDTA. NMR chemical shifts for the titration presented in *Figure 1—figure supplement 1* were collected using purified, recombinant $^{15}$N-labeled IST1$_{314-343}$ and unlabeled CAPN7$_{1-75}$, each in NMR buffer. Uniformly $^{15}$N-labeled IST1$_{314-343}$ was held constant at 200 µM while CAPN7$_{1-75}$ was titrated to 1 mM in 12 steps. The titration data were used to (1) map the CAPN(MIT)$_2$ binding site on IST1$_{303-366}$ (*Figure 1D*) and (2) quantify the equilibrium dissociation constant for complex formation (*Figure 1—figure supplement 1*), as described below.

To map the CAPN(MIT)$_2$ binding site on IST1$_{303-366}$, peak intensity ratios (unbound/bound) were measured for 55 main chain IST1$_{303-366}$ amide resonances (*Figure 1C and D* and *Figure 1—figure supplement 2*). A subset of 20 IST1 residues displayed large (>15-fold) changes in peak intensity upon saturation with CAPN7(MIT)$_2$ (binding average intensity ratio = 25 ± 7). These residues mapped exclusively to the two known IST1 MIMs. Most of these peaks lost intensity in a stepwise manner with each addition of CAPN7(MIT)$_2$ and were entirely absent upon saturation binding, consistent with slow

exchange kinetics, whereas a few displayed the loss of all intensity with sub-stochiometric amounts of CAPN7(MIT)$_2$, consistent with intermediate exchange kinetics (*Figure 1C* and *Figure 1—figure supplement 2*). The remaining IST1 backbone residues displayed small (<5-fold, 30 residues, average intensity ratio = 2.7 ± 0.8) or intermediate (5–15-fold, five residues, average intensity ratio = 6 ± 1) peak intensity ratio changes upon titration and therefore likely did not stably contact CAPN7(MIT)$_2$ or become well-ordered upon binding.

To quantify the dissociation constant for the CAPN7(MIT)$_2$-IST1$_{314\text{-}343}$ complex, we identified five assigned IST1 residues that displayed fast-exchange characteristics on titration with unlabeled CAPN7(MIT)$_2$ and were appropriate for equilibrium curve-fitting: F323, L325, T337, A338, and A340. A global fit was performed on the separate $^1$H and $^{15}$N chemical shifts for each residue (excluding $^{15}$N shifts for A338 and A340 because $^{15}$N Δδ were too small to reliably fit) to generate an average K$_D$ ± standard error of the mean from eight separate nuclei per titration point (*Figure 1—figure supplement 1B*, top panel). Binding isotherms for individual residues (*Figure 1—figure supplement 1B*, bottom panels) display combined $^1$H and $^{15}$N nuclei shifts according to the equation: $\delta_{combined} = (\Delta H^2 + (\Delta N/6.5)^2)^{1/2}$ (*Skalicky et al., 2012*; *Hobbs et al., 2022*). Isotherms in both cases were fit to the standard equation for 1:1 equilibrium binding (*Williamson, 2013*; *Hobbs et al., 2022*): Y = (0.5/P * Bmax)*((P + X + K$_D$) - (((P + X + K$_D$)$^2$) - (4 * P * X))$^{1/2}$).

## X-ray crystallography

In preparation for crystallization, purified CAPN7$_{1\text{-}165}$ (~60 mg/mL) and IST1$_{322\text{-}366}$ (~20 mg/mL), in a buffer containing 25 mM Tris (pH 7.2) at 23°C, 150 mM NaCl, 1 mM TCEP, 0.5 mM EDTA, were mixed in a 1:2 molar ratio (CAPN7:IST1) so that the final concentration of CAPN7$_{1\text{-}165}$ was 20 mg/mL. Crystallization was carried out in vapor-diffusion crystallization trays with the aid of an ARI Gryphon liquid-handling robot (Art Robbins Instruments). Crystals grew as flat hexagonal plates measuring 100–200 µm in largest dimensions after about 3 wk at 21°C in the Rigaku Wizard Cryo screen condition D5 (25% [v/v] 1,2-ropanediol, 100 mM sodium phosphate dibasic/citric acid pH 4.2, 5% [w/v] PEG 3000, 10% [v/v] glycerol). In preparation for data collection, crystals were transferred briefly (less than 20 s) into mother liquor with 25% added glycerol, suspended in a small nylon sample mounting loop, and cryocooled by plunging into liquid nitrogen.

X-ray diffraction data were collected at the Stanford Synchrotron Radiation Lightsource (SSRL). During data collection, the crystal was maintained at 100 K with the aid of a cold nitrogen gas stream. Data were integrated and scaled using XDS (*Kabsch, 2010a*; *Kabsch, 2010b*) and AIMLESS (*Evans, 2011*; *Evans and Murshudov, 2013*; *Table 1*). Initial phases were obtained using phaser in the PHENIX software suite (*Bunkóczi et al., 2013*) using VPS4B MIT (PDB 4U7Y) (*Guo and Xu, 2015*) as a search model. The resulting electron density maps were readily interpretable, allowing a model to be built using Coot (*Emsley and Cowtan, 2004*; *Emsley et al., 2010*), and refined with phenix.refine (*Liebschner et al., 2019*).

Model validation of the CAPN7 MIT domains was performed by generating an Fo-Fc map that demonstrated unbiased density for Phe and Tyr residues when these side chains were omitted from the model prior to refinement and calculation of model-based phases. Model validation of the IST1 MIM elements was performed by omitting all IST1 residues (*Figure 2—figure supplement 1*). The final model was refined against all data to R$_{work}$ = 0.211 and R$_{free}$ = 0.285. Full refinement statistics and details can be found in *Table 1*. Structure coordinates and diffraction data have been deposited in the RCSB Protein Data Bank (PDB 8UC6).

Structure alignments shown in *Figure 2* were generated using lsqkab (*Kabsch, 1976*) in the CCP4 program suite (*Winn et al., 2011*). Protein interfaces and details of protein-protein contacts were analyzed with PISA (*Krissinel and Henrick, 2007*) and LigPlot+ (*Laskowski and Swindells, 2011*).

## Gel filtration chromatography binding assays

Highly pure individual proteins were chromatographed by gel filtration at 4°C by injecting 2 nmol protein in a 100 µL sample loop onto a Superdex 200 column (24 mL; 10/300 GL, Cytiva Life Sciences) using 20 mM Tris (pH 7.2 at 23°C), 150 mM NaCl, 0.5 mM TCEP as the binding and eluent buffer (0.5 mL/min flow rate). Protein elution was monitored by UV absorbance at 280 nm. The column was calibrated using molecular weight standards (Bio-Rad).

Protein complexes were allowed to form by combining the two components in a 1:1 molar ratio to a final concentration of 20 µM in binding buffer and incubating on ice for 1 hr. Protein complexes were clarified by centrifugation at 16,000 × *g* for 10 min at 4°C, and 2 nmol of the complexes were immediately injected onto the column. Elution fractions were collected and the fraction corresponding to the CAPN7-IST1 peak was analyzed by Coomassie-stained SDS-PAGE to produce the inset image in *Figure 3B*.

## Size-exclusion chromatography with multiangle light scattering detection (SEC-MALS)

SEC-MALS analyses of purified proteins were performed at 23°C using a 24 mL Superdex 200 10/300GL connected to a Bio-Rad NGC chromatography system with miniDAWN and OptiLab detectors (Wyatt Technologies). Then, 50 µL of IST1 (50 µM) or CAPN7 (20 µM) were injected using a 100 µL sample loop, with 20 mM Tris (pH 7.2 at 23°C), 100 mM NaCl, 0.5 mM TCEP as eluent (at 0.5 mL/min). Molecular masses of peak fractions were estimated using ASTRA software (Wyatt Technologies).

## Cell culture

HEK293T and HeLa cells were cultured at 37°C and 5% $CO_2$ in high glucose DMEM (Gibco) supplemented with 10% FBS. TetOn-HeLa cells were additionally supplemented with 500 µg/mL G418 (Invitrogen) to maintain expression of the Tet-On Advanced protein. Doxycycline-inducible CAPN7 expression construct cell lines were generated in the parental Tet-On Advanced cell line and further supplemented with 0.5 µg/mL puromycin (Invitrogen).

## Cell lines

Cell lines were authenticated, generated, and validated as described previously (*Wenzel et al., 2022*). Briefly, the parental HeLa cell lines were authenticated by genomic sequencing of 24 loci (University of Utah Sequencing Core) and confirmed to be mycoplasma-free by routine PCR testing (ABM) following the manufacturer's protocol. To generate stable cell lines with doxycycline-inducible expression, the parental HeLa Tet-On Advanced cells were transduced with pLVX-tight puro lentiviral vectors containing the CAPN7 constructs (see *Supplementary file 2*) and selected for 14 d in 500 µg/mL G418 + 1 µg/mL puromycin. Single colonies were expanded and screened for CAPN7 expression by immunofluorescence and western blotting, with protein expression induced by addition of 1–2 µg/mL doxycycline. Selected clones were further validated by sequencing the PCR amplified exogenous CAPN7 construct.

## Co-immunoprecipitation experiments

HEK293T cells were seeded at 0.5 × 10⁶ cells per well in 6-well plates and transfected 24 hr later when cell confluency was ~80% with 3 µg DNA complexed with PEI. DNA mixtures contained 1.5 µg of pCAG plasmids encoding either Myc-IST1(190–366) or Myc-IST1(1–366), together with 1.5 µg of either empty vector (pCAG-MCS2-OSF) or the pCAG-CAPN7 construct (see *Supplementary file 2*). Cells were harvested 48 hr post transfection and lysed in 500 µL cold lysis buffer (50 mM Tris [pH 7.2 at 23°C], 150 mM NaCl, 0.5% TritonX-100, 1 mM DTT) supplemented with mammalian protease inhibitor cocktail (1:100, Sigma) by incubation for 15 min on ice with brief vortexing every 5 min. Lysates were clarified by centrifugation at 16,000 × *g* for 10 min at 4°C, then incubated with 20 µL of a 50% slurry of Streptactin resin (IBA Biosciences) for 1 hr at 4°C. The resin was washed three times with 500 µL lysis buffer and aspirated to near dryness. Bound proteins were eluted by boiling the Streptactin resin in 40 µL of 2× Laemmli sample buffer, resolved by SDS-PAGE, and detected by western blotting.

## siRNA transfections

For experiments shown in *Figure 4*, transfection protocols were as follows: day 1 – 0.5 × 10⁵ cells were seeded on fibronectin-coated coverslips in a 24-well plate for immunofluorescence, or 3 × 10⁵ cells in a 6-well plate for western blotting, and transfected with 20 nM siCAPN7 and 10 nM siNup153 complexed with Lipofectamine RNAiMAX (Invitrogen) (see *Supplementary file 3* for sequences) in media containing 2 µg/mL doxycycline. Then, 8 hr later, the media was changed for media that contained 2 µg/mL doxycycline and 2 mM thymidine. Day 2 – 24 hr later, the thymidine was washed

out with three washes of PBS and fresh media with 2 µg/mL doxycycline was added. Day 3 – cells were harvested 16 hr after thymidine washout for immunofluorescence and immunoblotting.

For experiments in *Figure 5A*, transfection protocols were performed with 72 hr siRNA knockdowns and 48 hr doxycycline-induced CAPN7 construct expression: day 1 – $5 \times 10^5$ cells were seeded in a 6-well plate and transfected with 20 nM siNT or siCAPN7 complexed with RNAiMAX (see *Supplementary file 3* for sequences). Day 2 – 24 hr later, cells were split onto either fibronectin-coated coverslips ($0.5 \times 10^5$ cells) for immunofluorescence or a 6-well plate for immunoblotting ($2.5 \times 10^5$ cells) and transfected again with 20 nM siNT or siCAPN7 in media containing 1 µg/mL doxycycline. Day 3 – 24 hr later, media was changed with media containing 1 µg/mL doxycycline. Day 4 – 24 hr later, cells were harvested for immunofluorescence and immunoblotting.

For experiments in *Figure 5C*, transfection protocols were performed with 72 hr siRNA knockdown of CAPN7 and 48 hr knockdown of Nup153 to induce a checkpoint arrest and 72 hr doxycycline-induced CAPN7 construct expression: day 1 – $5 \times 10^5$ cells were seeded in a 6-well plate and transfected with 20 nM siNT or siCAPN7 complexed with RNAiMAX with 2 µg/mL doxycycline. Day 2 – 24 hr later, cells were split onto coverslips or 6-well plate as above with 20 nM siNT or siCAPN7 plus 10 nM siNT or siNup153 with 2 µg/mL doxycycline. Day 3 – 24 hr later, media was changed with 2 µg/mL doxycycline. Day 4 – 24 hr later, cells were harvested for immunofluorescence and immunoblotting.

For experiments in *Figure 6A and B*, HeLa Tet-On Advanced cells were transfected twice with siRNAs for a total of 72 hr of knockdown. Day 1 – $5 \times 10^5$ cells were seeded in a 6-well plate and transfected with 20 nM siNT, siKATNA1, siSPAST, or siCAPN7, complexed with RNAiMAX (see *Supplementary file 3* for sequences). Day 2 – 24 hr later, cells were split onto either fibronectin-coated coverslips ($0.5 \times 10^5$ cells) for immunofluorescence or a 6-well plate for immunoblotting ($2.5 \times 10^5$ cells) and transfected again with 20 nM siRNA. Day 3 – changed media with 80 nM ICRF-193 (Sigma) or 0.4 µM aphidicolin (Sigma) or DMSO (negative control treatment). Day 4 – 24 hr later, cells were harvested for immunofluorescence and immunoblotting.

## Immunoblotting

Immunoblots related to *Figures 4 and 5* were performed as previously described (*Wenzel et al., 2022*). Briefly, cells were lysed in RIPA buffer (Thermo Fisher) supplemented with mammalian protease inhibitor cocktail (1:100, Sigma) for 15 min on ice with brief vortexing every 5 min. Lysates were cleared by centrifugation at $17,000 \times g$ for 10 min at 4°C. Lysate protein concentrations were determined by BCA Assay (Thermo Fisher), and 10 µg lysate per sample was prepared with SDS loading buffer, resolved by SDS-PAGE, and transferred to PVDF membranes. Membranes were blocked for 1 hr at room temperature in 5% milk in TBS, then incubated overnight at 4°C with primary antibodies (see *Supplementary file 4* for dilutions). Following $3 \times 10$ min washes in TBS-T, membranes were incubated with the appropriate secondary antibodies for 1 hr at 23°C, washed again with TBS-T, and imaged using a LiCor Odyssey infrared scanner.

## Immunofluorescence imaging and phenotype quantification

Cells were seeded on fibronectin-coated glass coverslips in 24-well plates and treated with siRNAs as described above. Coverslips were harvested by washing once with PBS and once with PHEM (25 mM HEPES, 60 mM PIPES, 10 mM EGTA, 4 mM $MgCl_2$, pH 6.9), then fixed and permeabilized in 4% PFA, PHEM, 0.5% TritonX-100 for 15 min at 23°C. Coverslips were then washed three times with PBS-T (0.1% TritonX-100) and blocked for 1 hr in 5% normal donkey serum in PBS-T. Coverslips were then incubated with primary antibody for 1–2 hr at 23°C, diluted in 1% BSA in PBS-T (see *Supplementary file 4* for antibody dilutions). Coverslips were then washed three times with PBS-T, then incubated for 1 hr at 23°C with secondary antibodies diluted in 1% BSA in PBS-T (see *Supplementary file 4* for antibody dilutions). Coverslips were washed three times with PBS-T, once with water, then mounted in ProLong Diamond (Invitrogen) for localization experiments (*Figure 4*) or ProLong Diamond with DAPI for functional experiments (*Figure 5*). Mounted coverslips were cured for at least 24 hr before imaging.

Images for *Figure 4A* were acquired on a Leica SP8 white light laser confocal microscope using a $63 \times 1.4$ oil HC PL APO objective. Images were acquired as Z-stacks and presented as maximum intensity Z-projections using the Leica App Suite X Software.

Images for *Figures 4B, 5A, C, 6A and B* phenotype quantification were acquired as previously described (*Wenzel et al., 2022*). Briefly, images were acquired using a Nikon Ti-E inverted microscope equipped with a ×60 PlanApo oil immersion objective, an Andor Zyla CMOS camera, and an automated Prior II motorized stage controlled with the Nikon Elements software. For phenotype quantification in *Figures 4–6*, the software was used to acquire 25 images using a randomized 5 × 5 grid pattern. The images were then blinded and scored to reduce any potential for bias. For *Figure 4B* localization quantification, only midbodies with in-focus IST1 staining were scored for the obvious presence or absence of mCherry signal. Quantification for *Figures 4B, 5A, C, 6A and B* was performed from three independent, biological replicates (cells seeded and treated on different days). Quantification and statistical analyses were performed using GraphPad Prism 9.

## Acknowledgements

We thank Julia Brasch and Alina Guo for their assistance in performing SEC-MALS analyses and Courtney Dailley for assistance in purifying proteins and performing gel filtration chromatography. Oligonucleotides and peptides ($IST1_{316-343}$ and $IST1_{344-366}$) were synthesized and purified by HPLC by the DNA/Peptide Core at the University of Utah. We thank Mike Hanson for assistance. DNA sequencing was performed by the DNA sequencing core facility at the University of Utah. We thank Derek Warner and Michael Powers for assistance. We thank the Cell Imaging Core at the University of Utah for use of equipment (Nikon Ti-E and Leica SP8 microscopes) and Xiang Wang for guidance. Protein mass spectrometry analyses were performed by the Mass Spectrometry and Proteomics Core at the University of Utah. We thank Sandra Osburn for assistance. Mass spectrometry equipment was obtained through a Shared Instrumentation Grant 1 S10 OD018210 01A1. Our work was funded by grants from NIH 5R01GM112080 (WIS, KSU, CPH) and F31GM139318 (ELP). Use of the Stanford Synchrotron Radiation Lightsource, SLAC National Accelerator Laboratory, is supported by the U.S. Department of Energy, Offices of Science, Office of Basic Energy Sciences under contract no. DE-AC02-76SF00515. The SSRL Structural Molecular Biology Program is supported by the DOE Office of Biological and Environmental Research, and by the NIH, National Institute of General Medicine Sciences (including P30GM133894). The contents of this publication are solely the responsibility of the authors and do not necessarily represent the official views of NIGMS or NIH.

## Additional information

### Funding

| Funder | Grant reference number | Author |
|---|---|---|
| National Institutes of Health | 5R01GM112080 | Katharine S Ullman Christopher P Hill Wesley I Sundquist |
| National Institutes of Health | F31GM139318 | Elliott L Paine |

The funders had no role in study design, data collection and interpretation, or the decision to submit the work for publication.

### Author contributions

Elliott L Paine, Conceptualization, Data curation, Formal analysis, Funding acquisition, Investigation, Methodology, Writing – original draft, Writing – review and editing; Jack J Skalicky, Conceptualization, Data curation, Formal analysis, Investigation, Methodology, Writing – review and editing; Frank G Whitby, Data curation, Formal analysis, Investigation, Methodology, Writing – review and editing; Douglas R Mackay, Methodology, Writing – review and editing; Katharine S Ullman, Conceptualization, Funding acquisition, Project administration, Writing – review and editing; Christopher P Hill, Funding acquisition, Investigation, Project administration, Writing – review and editing; Wesley I Sundquist, Conceptualization, Funding acquisition, Investigation, Project administration, Writing – review and editing

## Author ORCIDs

Elliott L Paine (iD) https://orcid.org/0000-0001-5575-297X
Jack J Skalicky (iD) http://orcid.org/0000-0002-5450-0567
Frank G Whitby (iD) http://orcid.org/0000-0003-3511-2216
Katharine S Ullman (iD) http://orcid.org/0000-0003-3693-2830
Christopher P Hill (iD) http://orcid.org/0000-0001-6796-7740
Wesley I Sundquist (iD) https://orcid.org/0000-0001-9988-6021

## Decision letter and Author response

Decision letter https://doi.org/10.7554/eLife.84515.sa1
Author response https://doi.org/10.7554/eLife.84515.sa2

## Additional files

### Supplementary files

- Supplementary file 1. ESI-MS mass confirmation of purified proteins.
- Supplementary file 2. Plasmids.
- Supplementary file 3. siRNA sequences.
- Supplementary file 4. Antibodies.
- MDAR checklist

### Data availability

X-Ray diffraction data were deposited in the PDB under accession code 8UC6. NMR chemical shift assignments for IST1(303-366) are available at Biological Magnetic Resonance Bank (Accession no: 25393; *Caballe et al., 2015*). All new plasmids generated for this study have been deposited at Addgene.

The following dataset was generated:

| Author(s) | Year | Dataset title | Dataset URL | Database and Identifier |
|---|---|---|---|---|
| Paine EL, Skalicky JJ, Whitby FW, Mackay DR, Ullman KS, Hill CP, Sundquist WI | 2022 | Calpain-7:IST1 complex | https://www.rcsb.org/structure/8UC6 | RCSB Protein Data Bank, 8UC6 |

The following previously published dataset was used:

| Author(s) | Year | Dataset title | Dataset URL | Database and Identifier |
|---|---|---|---|---|
| Dawn W, Jack S, Wesley S | 2015 | Backbone 1H, 13C, and 15N Chemical Shift Assignments for IST1 residues 303-366 | https://doi.org/10.13018/BMR25393 | BMRB, 10.13018/BMR25393 |

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
