## [Editor Report]

This fundamental study provides insight into the role of the Calpain-7 protease and its proteolytic activity in cell division. Using rigorous molecular and cellular approaches, they provide compelling evidence for the role of the protease in the timing and completion of cell abscission. Conclusions are supported with strong mutagenesis and rescue assays. The work will be of broad interest to cell biologists and biochemists.

---

## [Decision Letter]

**Decision letter after peer review:**

Thank you for submitting your article "The Calpain-7 protease functions together with the ESCRT-III protein IST1 within the midbody to regulate the timing and completion of abscission" for consideration by *eLife*. Your article has been reviewed by 3 peer reviewers, and the evaluation has been overseen by a Reviewing Editor and Vivek Malhotra as the Senior Editor. The following individual involved in the review of your submission has agreed to reveal their identity: Siavash Vahidi (Reviewer #1).

Essential revisions:

(1) The authors need to discuss the binding synergy they identified between the two MIM domains in depth. Is this binding cooperativity important for cell function? Does it play a role in the polymerization of ESCRT-II into filaments?

(2)The authors should include more details regarding NMR resonance assignments. Also, the authors should provide an additional figure showing the resonances that broaden or shift in the complex and map them onto the crystal structures.

3) Include raw data for FPA.

(4) Figure 2 – supplement 3 and the last two paragraphs of the NMR section would be considerably strengthened if dissociation constants from an NMR titration were provided.

(5) Indicate calibration on the SEC to assist with proper interpretation of the SEC data.

*Reviewer #1 (Recommendations for the authors):*

–– FPA is one of the main tools used in this paper and the authors do show several binding isotherms for the constructs they investigated. They should also include a few examples of raw data.

– How exactly were the FPA data fit? The authors state that they used the Prism software package for data fitting. They should show the exact equation(s) used and should describe specifically what the uncertainties displayed in parenthesis represent, how they were estimated, and using what methods (e.g. Monte Carlo simulations, jackknife resampling, etc.). Based on the binding isotherms and the goodness of the fits, I would have expected a higher degree of uncertainty in the dissociation constants derived. For example, for titrations that showed weak affinity, the authors should describe what exactly they are reporting, and the error associated with such extracted parameters (an upper bound? 1x standard deviation?).

– On page 2, last paragraph of the NMR section: The authors state they measured different chemical shifts for 23 amide resonances (this needs to be better defined, see comment below) in the MIT-bound state. This statement is not supported by the data. In Figure 1D, no chemical shift changes in IST were observed upon the addition of unlabeled MIT. The authors can only claim that IST peaks disappeared. It is unlikely that the chemical shifts stayed the same, but since the bound peaks are not observed, they have no measurement to support the magnitude of the chemical shift changes (if any).

– Figure 1D shows changes in peak heights and not chemical shift perturbations. Calling the y-axis chemical shift perturbations (when none/few are measured) and assigning them values of -1 and +1 is unusual and confusing. I suggest modifying this figure and the text above so that the data are discussed in terms of peak intensities, not chemical shifts (see note above). A simple solution would be to plot intensity ratios between the bound and free spectra on the vertical axis. An inferior yet acceptable solution would be to set the intensity to 1 for peaks that stay visible and set it to 0 for those that disappear. In either case, the y-axis should be relabeled. The overall outline of Figure 1D can stay the same. The protein sequence and schematic illustrating MIM locations are done well and should stay as is.

– In the text, I would suggest that authors state that for 23 residues the peaks disappear, for X number of peaks (where X<31) peak intensities change to some percentage range (peak heights in free and bound IST1 will be different), and perhaps for Y number of peaks (Y=31-X) intensities could not be quantified due to overlap or other reasons. The larger than expected size of the interacting proteins could explain why the peaks in the NMR spectra disappear in the bound state (see my comment on complex stoichiometry below).

– In the crystallography section, the authors state that the asymmetric unit contains two copies of the MIT-IST complex and outline two possible ways of modeling their structure. They clearly state that they chose to use the simpler model, but I believe showing the alternative arrangement as a supplementary figure would add value to the manuscript.

– Compared to FPA, NMR is a far better technique for quantifying weak interactions between individual MIMs and MIPs. Figure 2 – supplement 3 and the last two paragraphs of the NMR section would be considerably strengthened if dissociation constants from an NMR titration were provided.

– In Figure 3B, the authors use size exclusion chromatography (SEC) to claim that a 1:1 complex is formed between MIT and IST. This is by no means supported by the data they provide. The authors mention that they calibrated their Superdex 200 column, but calibration data is not provided. We happen to own the same type of SEC column in our lab. In our hands, a 100 kDa protein elutes at ~13.3 mL retention on our column. Figure 3B indicates that CAPN7 is a monomer, but IST1 elutes earlier, consistent with it being a dimer or even a trimer. The retention volume of the wild-type complex appears more consistent with a 2:2 complex rather than a 1:1.

– The calibration data should be added to the figure. Regardless, I highly recommend the stoichiometry of this complex and isolated IST and MIT domains needs to be validated using another measurement such as NMR- or DLS-based measurement of diffusion rates, SEC-MALS (recommended), or native mass spectrometry. Establishing the stoichiometry of the complex and the interacting proteins is especially important because it might affect the models used in the fitting of the FPA data and their reported dissociation constants, the interpretation of the X-ray data, and the overall mechanism of ESCRT cytokinesis.

-Figure 5A: The style of the stacked bar chart makes this figure hard to read. I strongly recommend the authors improve this. One solution would be to change the chart to two bars per treatment (which has its own issue with visual clutter). Alternatively, they could separate midbody vs. multinucleate data into panels C and D (which would be my preference).

*Reviewer #2 (Recommendations for the authors):*

I have the following recommendations:

1. The sentence starting on page 4, lines 16 to 18 needs a reference or a supplementary figure showing the sequence divergence.

2. Bottom of p. 4 and top of p. 5, please define type-2 and type-3 interactions.

3. Page 5, line 23 "The crystal structure of the CAPN7(MIT)2 – IST1322-366 complex explains why IST1 binding is disrupted by the V18D mutation in the first 25 CAPN7 MIT domain…" explain why?

4. Figure 3C – IP: Streptactin: using the unedited source data in the main figure would be more suitable. The source data shows "that the full-length Myc-IST1 showed modest levels of background binding" – why is this background binding? It could as well represent functionally relevant interaction.

*Reviewer #3 (Recommendations for the authors):*

Before publication, the authors need to address the following questions/comments:

(A) The authors need to discuss the binding synergy they identified between the two MIM domains in depth. Is this binding cooperativity important for cell function? Does it play a role in the polymerization of ESCRT-II into filaments?

(B) The authors should include more details regarding NMR resonance assignments. Also, the authors should provide an additional figure showing the resonances that broaden or shift in the complex and map them onto the crystal structures.

---

## [Author Response]

Essential revisions:(1) The authors need to discuss the binding synergy they identified between the two MIM domains in depth. Is this binding cooperativity important for cell function? Does it play a role in the polymerization of ESCRT-II into filaments?

The binding synergy between the two CAPN7 MIT domains and their target IST1 MIM elements is a central theme of our paper. Specifically, we show that both interaction interfaces contribute to the CAPN7-IST1 interaction in vitro and in cells (Figure 3), and quantify the energetic contributions of each interface (Figure 3A). Both interaction interfaces are also required for proper localization of CAPN7 to midbodies (Figure 4) and for CAPN7 functions in abscission and NoCut checkpoint regulation (Figure 5). We appreciate that the reviewer highlighted this concept, and we have included text that emphasizes the importance of both interaction interfaces, e.g., discussions of Figure 3 (page 6, lines 8-9 and lines 19-21), Figure 4 (page 7, lines 5-7), and Figure 5 (page 7, lines 21-22 and lines 31-32).

On the question of the importance of the cooperating MIM interactions contributing to ESCRT-III filament polymerization: we have presented strong evidence that the interaction between CAPN7 and IST1 does NOT play a role in ESCRT-III polymerization because IST1 ESCRT-III polymers assemble in vitro in the absence of CAPN7, and also in the absence of the C-terminal region of IST1 (which contains the MIM elements that bind cooperatively to CAPN7) (PMID: 26634441). Furthermore, our immunofluorescence images show that localization of IST1 polymers in the midbody is not affected by CAPN7 depletion, or by CAPN7 MIT mutants that inhibit IST1 binding to either the first (V18D) or second (F98D) MIT domains (see manuscript Figure 4A, and quantified in Figure 4B). Therefore, ESCRT-III polymers assemble in vitro and in cells without IST1 MIM-CAPN7 MIT interactions (and, indeed, even in the absence of CAPN7).

(2)The authors should include more details regarding NMR resonance assignments. Also, the authors should provide an additional figure showing the resonances that broaden or shift in the complex and map them onto the crystal structures.

In response to these helpful suggestions, we have (1) added text to our Materials and methods section (page 22, lines 49-52 through page 23 lines 1-4; “NMR data were collected on a Varian INOVA 600 MHz spectrometer running OpenVnmrJ_v3.1. 2D ^15^N-HSQC data were processed using nmrPipe (Delaglio *et al.*, 1995) and spectra were analyzed and NMR figures prepared using NMRFAM-SPARKY (Lee *et al.*, 2015). NMR resonance assignments for ^15^N-labeled IST1_303-366_ were taken from Caballe *et al.*, 2015. Briefly, sequential assignments were obtained from 2D ^15^N-HSQC and 3D HNCA, HNCACB, CBCAcoNH, HNCO, and HNcaCO triple resonance experiments. Using this approach, it was possible to assign all 55 non-proline backbone amide resonances. Chemical shift assignments were deposited in the BMRB (Accession no: 25393).”), and (2) updated (Figure 1D) with display of IST1 intensity changes induced by

CAPN7(MIT)_2_ binding and make comparison of NMR mapped MIM boundaries vs. X-ray defined MIM boundaries. These will allow readers to compare the NMR and crystal structure data more easily (and see that they are in good agreement).

(3) Include raw data for FPA.

In response to this helpful suggestion, we have created a new supplemental figure (Figure 1 —figure supplement 4) that shows the raw data and best-fit statistics for each of three replicate binding isotherms corresponding to the FPA data presented in Figure 1B. These data are illustrative because the three isotherms range from a high affinity interaction with a high confidence K_D_ to a weak interaction with an undetermined K_D_. This data will allow readers to evaluate the quality of our raw FPA binding data.

(4) Figure 2 – supplement 3 and the last two paragraphs of the NMR section would be considerably strengthened if dissociation constants from an NMR titration were provided.

In response to this helpful suggestion, we performed the requested titration experiment and now present the results in the new Figure 1 —figure supplement 1. Titration data for five shifted, fast-exchanging residues were analyzed individually (the binding isotherms for three such resonances, F323, L325, and T337, are shown), and also globally fit to a simple 1:1 binding model (top right panel), providing a dissociation constant of 200 ± 20 µM, which is in excellent agreement with our FPA estimate that the K_D_ is modestly above 100 µM.

(5) Indicate calibration on the SEC to assist with proper interpretation of the SEC data.

In response to this helpful suggestion, we have now added the positions of SEC calibration standards directly above the aligned chromatograms (Figure 3B) to allow readers to see the elution positions of the SEC calibration standards. We have also added a new supplemental figure that presents the raw chromatograms of the calibration data and individual protein traces (Figure 3—figure supplement 2A) so that reviewers can assess the data quality. Most importantly, as described in greater detail in our response to Reviewer #1, we have now performed SEC-MALS analyses on the individual CAPN7 and IST1 proteins to show that they both elute as monomeric proteins. We have included these data in new Figure 3—figure supplement 2B. These data have further confirmed our original assignments and interpretations.

Reviewer #1 (Recommendations for the authors):–– FPA is one of the main tools used in this paper and the authors do show several binding isotherms for the constructs they investigated. They should also include a few examples of raw data.

We have done this, as described above in “Essential Revisions,” Point 3.

– How exactly were the FPA data fit? The authors state that they used the Prism software package for data fitting. They should show the exact equation(s) used and should describe specifically what the uncertainties displayed in parenthesis represent, how they were estimated, and using what methods (e.g. Monte Carlo simulations, jackknife resampling, etc.). Based on the binding isotherms and the goodness of the fits, I would have expected a higher degree of uncertainty in the dissociation constants derived. For example, for titrations that showed weak affinity, the authors should describe what exactly they are reporting, and the error associated with such extracted parameters (an upper bound? 1x standard deviation?).

In response to this helpful suggestion, we have added more detailed information on how the FPA data were fit (page 22, lines 37-46). As noted above, we also have included raw FPA binding data for the isotherms in Figure 1B, together with associated curve-fits and statistics so the reviewer and readers can judge the quality of the raw data. We have also updated the figure legends and Materials and methods to state more clearly that reported K_D_’s are the averages ± standard error of mean from three independent measurements. For weak affinity binding (in cases where the fraction bound did not reach 0.5), we did not attempt to estimate K_D_’s, but instead reported the K_D_ as being greater than the highest concentration point measured (e.g., in Figure 1B, binding of IST1_314-343_ to CAPN7(MIT)_2_ is reported as K_D_ > 100 µM because the binding isotherm shows only ~30% saturation at that CAPN7(MIT)_2_ concentration). Figure legends and Materials and methods have been updated to clarify these points.

– On page 2, last paragraph of the NMR section: The authors state they measured different chemical shifts for 23 amide resonances (this needs to be better defined, see comment below) in the MIT-bound state. This statement is not supported by the data. In Figure 1D, no chemical shift changes in IST were observed upon the addition of unlabeled MIT. The authors can only claim that IST peaks disappeared. It is unlikely that the chemical shifts stayed the same, but since the bound peaks are not observed, they have no measurement to support the magnitude of the chemical shift changes (if any).– Figure 1D shows changes in peak heights and not chemical shift perturbations. Calling the y-axis chemical shift perturbations (when none/few are measured) and assigning them values of -1 and +1 is unusual and confusing. I suggest modifying this figure and the text above so that the data are discussed in terms of peak intensities, not chemical shifts (see note above). A simple solution would be to plot intensity ratios between the bound and free spectra on the vertical axis. An inferior yet acceptable solution would be to set the intensity to 1 for peaks that stay visible and set it to 0 for those that disappear. In either case, the y-axis should be relabeled. The overall outline of Figure 1D can stay the same. The protein sequence and schematic illustrating MIM locations are done well and should stay as is.

The reviewer is correct, and we have remade Figure 1D and changed our descriptions of the NMR data to note that our mapping measurements reflect changes in resonance intensities (not chemical shift changes).

– In the text, I would suggest that authors state that for 23 residues the peaks disappear, for X number of peaks (where X<31) peak intensities change to some percentage range (peak heights in free and bound IST1 will be different), and perhaps for Y number of peaks (Y=31-X) intensities could not be quantified due to overlap or other reasons. The larger than expected size of the interacting proteins could explain why the peaks in the NMR spectra disappear in the bound state (see my comment on complex stoichiometry below).

We now explain in the main text that 20 IST1 resonances had large intensity changes upon CAPN7(MIT)_2_ binding, and that these changes map exclusively to the two IST1 MIM elements (page 3, lines 5-14). We also explain in greater detail in the Materials and methods section (page 23, lines 16-28). There were not any resonances whose intensities could not be quantified due to overlap.

– In the crystallography section, the authors state that the asymmetric unit contains two copies of the MIT-IST complex and outline two possible ways of modeling their structure. They clearly state that they chose to use the simpler model, but I believe showing the alternative arrangement as a supplementary figure would add value to the manuscript.

We have created Author response image 1 for the reviewer’s information. We respectfully disagree that this figure adds value to the manuscript (since we clearly state that our data do not distinguish between the two possibilities and we chose what we consider to be the simpler of the two models, which we prefer because it minimizes linker crossovers and because the undefined linkers seem to follow the projected helical paths of MIT1 helix 3 and MIT2 helix 1).

**Author response image 1. sa2fig1:** Mock figure showing the structure model presented in the manuscript, overlaid with another possible model with alternative connectivity between the two copies of the complex within the crystallographic asymmetric unit. The two structures were aligned to the first MIT domain to highlight the divergence between possible interdomain linkers and C-terminal domain positions.

– Compared to FPA, NMR is a far better technique for quantifying weak interactions between individual MIMs and MIPs. Figure 2 – supplement 3 and the last two paragraphs of the NMR section would be considerably strengthened if dissociation constants from an NMR titration were provided.

We provided these, as described in our response to “Essential revisions”, Point 4.

– In Figure 3B, the authors use size exclusion chromatography (SEC) to claim that a 1:1 complex is formed between MIT and IST. This is by no means supported by the data they provide. The authors mention that they calibrated their Superdex 200 column, but calibration data is not provided. We happen to own the same type of SEC column in our lab. In our hands, a 100 kDa protein elutes at ~13.3 mL retention on our column. Figure 3B indicates that CAPN7 is a monomer, but IST1 elutes earlier, consistent with it being a dimer or even a trimer. The retention volume of the wild-type complex appears more consistent with a 2:2 complex rather than a 1:1.– The calibration data should be added to the figure. Regardless, I highly recommend the stoichiometry of this complex and isolated IST and MIT domains needs to be validated using another measurement such as NMR- or DLS-based measurement of diffusion rates, SEC-MALS (recommended), or native mass spectrometry. Establishing the stoichiometry of the complex and the interacting proteins is especially important because it might affect the models used in the fitting of the FPA data and their reported dissociation constants, the interpretation of the X-ray data, and the overall mechanism of ESCRT cytokinesis.

We appreciate the chance to clarify this issue. Firstly, we have now added elution positions for the SEC calibration standards to the main figure and also show the raw data in a supplemental figure (see “Essential Revisions”, Point 5 above). The reviewer is correct that IST1 elutes earlier than CAPN7 despite its smaller molecular weight, but this is due to its elongated shape (PMID: 18032582 and 19129479). Importantly, we have now performed SEC-MALS analyses on both individual proteins to show that both eluting peaks correspond to monomeric proteins. We have included these data in Figure 3—figure supplement 2B and added text to clarify this point on page 5, lines 32-35 through page 6, lines 1-4:

“We also assessed the importance of these interfaces for association of the full-length IST1 and CAPN7 proteins, using size exclusion chromatography (SEC) to analyze the individual proteins and their complex (Figure 3B and Figure 3figure supplement 2). SEC with multi-angle light scattering (SEC-MALS) analyses showed that the individual IST1 and CAPN7 proteins both eluted as monomers (IST1: calculated MW = 40.0 kDa, SEC-MALS estimated MW = 41 ± 1 kDa; CAPN7 calculated MW = 92.6 kDa, SEC-MALS estimated MW = 90.4 ± 0.3 kDa, see Figure 3—figure supplement 2B). However, IST1 eluted anomalously rapidly, apparently because it adopts an extended conformation.”

Finally, we also attempted to perform analogous SEC-MALS analysis of the complex but our data were not rigorously interpretable because the complex dissociates slightly during chromatography (and we are limited in the concentration of CAPN7 that we can achieve). Nevertheless, those data again indicate that the complex is also monomeric (i.e., 1:1) because the estimated SEC-MALS mass (105 kDa) is slightly *lower* than that expected for a 1:1 complex (130 kDa).

-Figure 5A: The style of the stacked bar chart makes this figure hard to read. I strongly recommend the authors improve this. One solution would be to change the chart to two bars per treatment (which has its own issue with visual clutter). Alternatively, they could separate midbody vs. multinucleate data into panels C and D (which would be my preference).

While we appreciate the reviewer’s suggestion, we feel it best to maintain this style to match our previous publication (which is also the most common presentation of these type of data in our field).

Reviewer #2 (Recommendations for the authors):I have the following recommendations:1. The sentence starting on page 4, lines 16 to 18 needs a reference or a supplementary figure showing the sequence divergence.

We thank the reviewer for pointing this edit out, and we have now added a sequence alignment to the bottom of Figure 2 —figure supplement 2A and added a reference to this figure in the text (page 4, line 12).

2. Bottom of p. 4 and top of p. 5, please define type-2 and type-3 interactions.

We have added clarifying language to these two paragraphs.

3. Page 5, line 23 "The crystal structure of the CAPN7(MIT)2 – IST1322-366 complex explains why IST1 binding is disrupted by the V18D mutation in the first 25 CAPN7 MIT domain…" explain why?

We now explain that “replacing either of these hydrophobic residues with charged residues would create unfavorable interactions in the hydrophobic cores of the MIT binding interfaces.” (page 5, lines 20-22).

4. Figure 3C – IP: Streptactin: using the unedited source data in the main figure would be more suitable. The source data shows "that the full-length Myc-IST1 showed modest levels of background binding" – why is this background binding? It could as well represent functionally relevant interaction.

The context for this experiment was to examine the effect of disrupting the C-terminal binding interfaces in the two MIM elements of IST1. We chose to include the C-terminal IST1 construct for the pulldown in Figure 3C because the C-terminal construct includes these interfaces and lacks the ability to oligomerize (PMID: 19525971), which often contributes to background binding. Nevertheless, we agree that the modest binding seen for full-length IST1 mutants could conceivably reflect minor binding contributions from other region(s) of the full-length protein. We have therefore included the raw binding data in a supplemental figure so that readers can see the data for themselves, and we now explain that the very low levels of binding for the full length could indicate “either non-specific background binding or minor binding contributions from other region(s) of the protein.” (page 6, lines 18-19).

Reviewer #3 (Recommendations for the authors):Before publication, the authors need to address the following questions/comments:(A) The authors need to discuss the binding synergy they identified between the two MIM domains in depth. Is this binding cooperativity important for cell function? Does it play a role in the polymerization of ESCRT-II into filaments?

We have answered these questions, as explained in response to “Essential revision”, Point 1.

(B) The authors should include more details regarding NMR resonance assignments. Also, the authors should provide an additional figure showing the resonances that broaden or shift in the complex and map them onto the crystal structures.

We have included the recommended information, as explained in response to “Essential revision”, Point 2.